# Neuroimaging evidence for a network sampling theory of individual differences in human intelligence test performance

Eyal Soreq [1] ✉, Ines R. Violante [2], Richard E. Daws[1] & Adam Hampshire[1]

Despite a century of research, it remains unclear whether human intelligence should be studied as one dominant, several major, or many distinct abilities, and how such abilities relate to the functional organisation of the brain. Here, we combine psychometric and machine learning methods to examine in a data-driven manner how factor structure and individual variability in cognitive-task performance relate to dynamic-network connectomics. We report that 12 sub-tasks from an established intelligence test can be accurately multi-way classified (74%, chance 8.3%) based on the network states that they evoke. The proximities of the tasks in behavioural-psychometric space correlate with the similarities of their network states. Furthermore, the network states were more accurately classified for higher relative to lower performing individuals. These results suggest that the human brain uses a high-dimensional network-sampling mechanism to flexibly code for diverse cognitive tasks. Population variability in intelligence test performance relates to the fidelity of expression of these task-optimised network states.

[1] The Computational, Cognitive and Clinical Neuroimaging Laboratory, Department of Brain Sciences, Faculty of Medicine, Imperial College London, London, UK. [2] School of Psychology, Faculty of Health and Medical Sciences, University of Surrey, Guildford, UK. ✉email: eyal.soreq@ukdri.ac.uk

The question of whether human intelligence is dominated by a single general ability, 'g'[1], or by a mixture of psychological processes[2–6], has been the focus of debate for over a century. While performance across cognitive tests does tend to positively correlate, population-level studies of intelligence have clearly demonstrated that tasks which involve similar mental operations form distinct clusters within a positive correlation manifold. These task clusters exhibit distinct relationships with various sociodemographic factors that are not observable when using aggregate measures of intelligence, such as 'g'[2,7].

Recent advances in network science offer the potential to resolve these contrasting views. It has been proposed that transient coalitions of brain regions form to meet the computational needs of the current task[8–10]. These dynamic functional networks are thought to be heavily overlapping, such that any given brain region can express flexible relationships with many networks, depending on the cognitive context[8,9,11–13]. This dynamic network perspective represents a major departure from localist models of brain functional organisation. Instead of cognitive functions mapping to discrete neural regions or specific connections, mental operations are suggested to be supported by unique conjunctions of distributed brain regions, en masse. The set of possible conjunctions can be considered as the repertoire of dynamic network states and the expression of these states may differ across individuals and relate to cognitive performance.

This conceptual shift motivates us to propose a network sampling theory of intelligence, which is conceptually framed by Thomson's classic sampling theory[14]. Thomson originally proposed that 'every mental test randomly taps a number of 'bonds' from a shared pool of neural resources, and the correlation between any two tests is the direct function of the extent of overlap between the bonds, or processes, sampled by different tests'. Extending this hypothesis, network sampling theory views the set of connections in the brain that constitute a task-evoked dynamic network state to be equivalent to Thomson's 'bonds'; therefore, the set of available brain regions is equivalent to the 'shared pool of neural resources'. The distinctive clusters within the positive manifold reflect the tendency of operationally similar tasks to rely on similar dynamic networks[2,15–17]. From this perspective, the general intelligence factor 'g' is proposed to be a composite measure of the brain's capacity to switch away from the steady state, as measured in resting-state analyses, in order to adopt information processing configurations that are optimal for each specific task. When recast in this framework, classic models of unitary and multiple-factorial intelligence[1,14] are reconciled as different levels of summary description of the same high-dimensional dynamic network mechanism. The notion of domain-general systems such as 'task active' or 'multiple-demand' cortex is also reconciled within this framework. Specifically, each brain region can be characterised by the diversity of network states they are active members of. Brain regions that classic mapping studies define as 'domain-general' place at one extreme of the membership continuum, whereas areas ascribed specific functions, e.g., sensory or motor, place at the other extreme. The aim of this study was to test key predictions of network sampling theory using 12 cognitive tasks and machine learning techniques applied to functional MRI (fMRI) and psychometric data. First, we test the hypothesis that cognitive tasks evoke distinct configurations of activity and connectivity in the brain. We predicted that these configurations would be sufficient to reliably classify individual tasks, and that this would be the case even when focusing on brain regions at the domain-general extreme of the network membership continuum. We then tested Thomson's hypothesis that similarity between cognitive tasks maps to the 'overlap' of the neural resources being tapped. Subsequently, it was predicted that the ability to classify pairs of tasks

would negatively correlate with their behavioural-psychometric similarity, with tasks that are less similar being classified more reliably. Next, we hypothesised that individual functional dynamic repertoires would positively correlate with task performance, with the top performers expressing task configurations that would be more reliably classified. We also tested the prediction that classification success rates should have a basis in a combination of the distinct visual (VS), motor and cognitive subprocesses of the tasks. Finally, we hypothesised that task performance would be associated with optimal perturbation of the network architecture from the steady state, and that certain features within the network would have more general and more prominent roles in intelligence test performance.

## Results

**Data scope.** We analysed fMRI data collected from 60 healthy young adults while they performed 12 sub-tasks of an established intelligence test, which has been previously used to assess 44,780 and 18,455 members of the general public in two internet-based studies[2,7]. During fMRI recordings, each task was performed in three 1-min blocks interleaved with 20 s of rest.

**Spatial overlap in domain-general activation.** We first sought to identify 'domain-general' regions of the brain, that is, the brain regions that were most consistently active across the 12 tasks. To do that, we started by creating a group average activation map for each of the 12 tasks (Fig. 1a). Using dice coefficients (DICE), we demonstrated that any pair of tasks exhibited activation patterns with high spatial similarity (mean DICE = 0.8, min DICE 0.7). We followed this by generating three statistical maps: (1) an intersection (INTR) map, which was calculated across all 12 tasks as the conjunction of voxels that are active for all tasks (Fig. 1b); (2) a T-contrast map comparing activity during the 12 tasks relative to rest (Fig. 1c) and (3) a logical union map (Fig. 1d), i.e. all voxels that are active for at least one task. To have a better idea of the level of overlapping across tasks, we quantified the multi-way percentage of task overlapping within the logical union map (Fig. 1e, f). This indicated that more than 50% of voxels were active in at least nine tasks, and that voxels that are purely domain-specific are located at the boundaries of the union map. Overall, the patterns observed in the statistical maps correspond to the previously reported 'multiple-demand' (MD) cortex[18–20], which includes frontal, parietal, VS and motor brain regions. This pattern is observed irrespective of the more extensive (union map) or restricted (intersection map) volume of activation. Furthermore, comparison of the INTR map to a previously published resting state network (RSN) atlas[21] showed a close correspondence to VS and attention networks, which in turn functionally sub-divide MD cortex[2,22].

**Functional heterogeneity within the domain-general to domain-specific continuum.** To more accurately quantify the level of functional variability within MD cortex, the INTR map was parcellated into regions of interest (ROIs) using an in-house watershed algorithm[12,23]. For comparison, two further atlases derived from independent studies were also examined. The first captured regions that typically activate (MD) and deactivate (default mode network (DMN)) across large sets of cognitive tasks, that we refer to as MDDM[20]. The second was a 200 ROI resting state functional parcellation[24] of the cortex (CRTX), i.e., ROI's that were defined independently of domain generality (see Fig. 2a–c bottom panel). Together, these ROI sets capture different mixtures of the domain-general to domain-specific continuum.

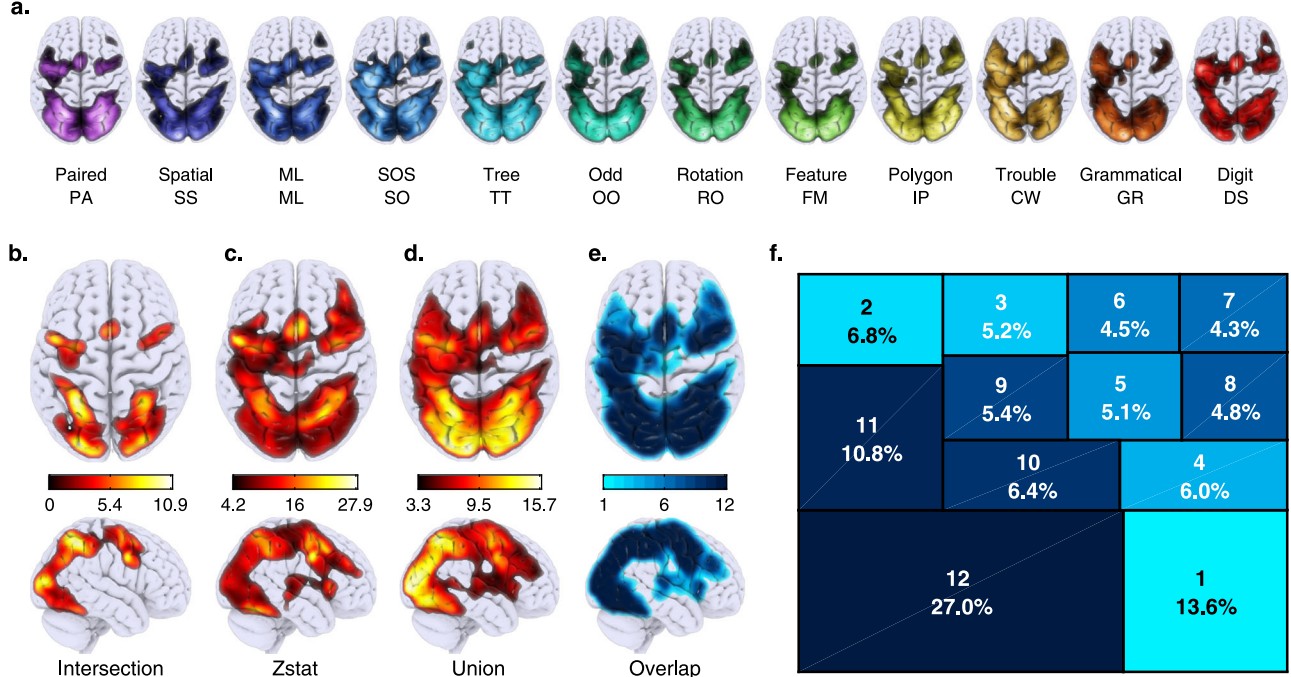

**Fig. 1 Domain-general overlap volumes.** We used a cognitive battery designed to measure different aspects of intelligence to collect task-evoked brain activation data from 60 participants. A description of each task is provided in Supplementary Methods A.1 and graphical representations in Fig. S1 and Supplementary movies 1–12. **a** Maximum intensity projection for the voxel-wise activation patterns (vs resting period) for the 12 cognitive tasks composing our battery. Initials for each task are displayed underneath simplified task names. **b–d** Maximum intensity projection for the following: **b** the intersection of the 12 tasks, i.e., voxels that were active for all of the tasks; **c** $t$-test across the 12 tasks relative to rest, i.e., tasks that were reliably active for multiple tasks. **d** Union of the 12 task activation patterns, i.e., voxels that were active for at least one task. **e** Proportion voxel-wise overlap projection (i.e. the number of tasks a voxel is activated by (darker colour = voxels present in more tasks). **f** Treemap quantifying the multi-way proportion of logical intersections of positive activation patterns across tasks. Top digit indicating the number of overlapping tasks and the bottom the percentage of voxels existing in the k-way intersection. Colour matches the overlap volume.

We used these three parcellations to extract voxel-wise vectors of activation for each task block and each subject and examined the distribution of activity across each parcellation (Fig. 2d). As expected, the INTR set included predominantly positive values and the MDDM set had a bimodal distribution, reflecting the inclusion of 'task-positive' and 'task-negative' brain regions. The CRTX set had a broad right-tailed distribution with a negative peak. Notably, these broad activation distributions were indicative of substantial variability in task-evoked responses even amongst the most commonly active INTR set of brain regions. To investigate this further, we generated voxel-wise activation distributions for each of the tasks and parcellations (Fig. 2e) and then calculated pairwise similarity matrices (estimated using cross-correlation see Fig. 2f) for each parcellation. To quantify the apparent clustering, the similarity matrices were analysed using principle components analysis (PCA), and this produced 3–4 components (Fig. 2f). Taken together, these analyses demonstrate that brain regions that are active across the 12 tasks exhibit activation patterns that are specific to individual, or clusters of, tasks. This accords well with the prediction of network sampling theory that cognitive tasks should sample different but heavily overlapping combinations of brain regions.

**Relating behavioural factor structure to neuronal overlap.** A central prediction of network sampling theory is that the strength of the behavioural psychometric correlations between tasks should correspond to the degree to which they tap common underlying neural resources. To define psychometric similarity, we used our previously published latent factor structure derived using >60k test scores from the general public[2,7] which loads the

12 tasks used here onto three orthogonal psychometric factors, visuospatial (VS), reasoning (RE) and verbal reasoning (VR) (Fig. 3a, b). As predicted by network sampling theory, the psychometric distance between each pair of tasks showed a strong positive correlation with the corresponding whole-brain task activation DICE coefficients ($r = 0.63$, $p < 0.001$, Fig. 3c, d). This indicates that tasks that are psychometrically similar evoke similar patterns of activation. This strong association at the whole-brain level held when separately testing each of the three ROI sets (Fig. 3e, $r_{INTR} = 0.54$, $r_{MDDM} = 0.60$, $r_{CRTX} = 0.60$, all $p < 0.001$).

**Twelve-way classification of cognitive tasks based on brain activation patterns.** We next engineered a multivariate classification pipeline to test whether the 12 tasks could be accurately identified based on the patterns of BOLD activation (BA) that each task evoked using voxel-wise activity from each of the three ROI sets. To eschew the over-fit problem, for each ROI set, we separated our data into two independent subject subsets (training = 75% and test = 25%). We used the training data to fit one true 12-way classification model and a null model using a scrambled response vector. Model performances were estimated within the training set using fivefold cross-validation (CV) and on the naïve test set (held-out). This process was repeated 100 times to form performance distributions. As expected, the CV estimates were overly optimistic[25], with all three ROI sets showing a significant 15% improvement in performance (estimated using $F1$-macro) compared to the held-out test sets ($p < 0.0001$). Consequently, we report the performance from the held-out test set in all subsequent analyses. The ROI voxel-wise BA patterns were

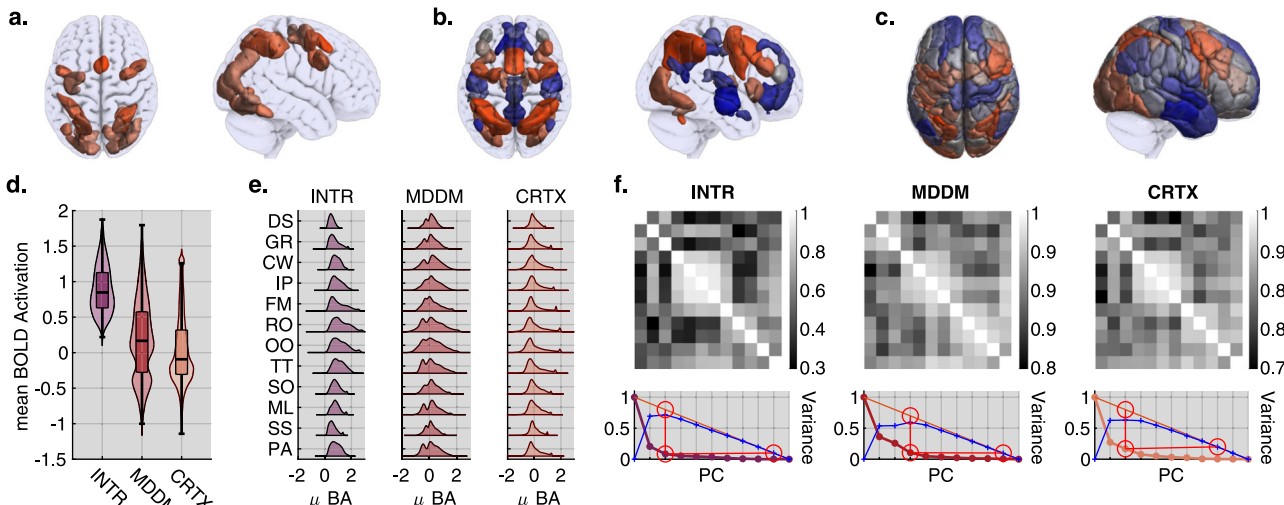

**Fig. 2 Functional heterogeneity within ROI sets. a–c** Top panel shows the region of interest parcellation sets colour coded to reflect task positive (orange) or negative (blue) associations quantified using mean activation for each ROI. **a** The intersection conjunction was segmented using our in-house watershed algorithm. **b** The task negative to positive[20] statistical volume was segmented using the same method. **c** Unbiased ROI set capturing the whole cortex as defined by[24] during resting state fMRI. **d** Using the above ROI sets we use custom box-violin plots to show the grand mean voxel-wise activation distribution contained within each brain mask (boxes height is defined as the inter quantile range (IQR); i.e. the range from the 3rd to the 1st quantile, the median is drawn as the central line. Lower inner fence is defined as the 1st quantile minus 1.5 times IQR; upper inner fence is defined as the 3rd quantile plus 1.5 times IQR. Finally, the violin plot is calculated from the winsorized kernel density distribution. Full code is avaliable in the GitHub repository under fws.plot.group_violin.m. As expected, the INTR contains only positive values, the MDDM has a bimodal distribution reflecting the task positive and negative networks it captures; and the CRTX mask has a skewed unimodal distribution spanning positive and negative values with a long right tail. **e** Task-wise distribution plots across ROI sets emphasise the heterogeneity of the different tasks. **f** Top panel shows the pairwise correlation coefficient matrices for each ROI set across tasks. Note, the inner clustering of tasks is consistent across ROI sets. **f** Bottom panel shows scree plots for principle component analysis (PCA) of task × ROI, separately for each ROI set across tasks. Note, the optimal number of principle components needed to explain the variance across tasks is at least three for each ROI set, thereby demonstrating the heterogeneity of their activation responses to different tasks. Source data are provided as a Source Data file.

sufficient in 12-way classifying the tasks, the mean classification accuracy across ROI sets was 34.5% (chance 8.3%). However, the classification accuracy varied across ROI sets (one-way repeated-measures ANOVA $F_{2,297} = 452$, $p < 0.0001$), with performance increasing as a function of the ROI set's spatial extent ($F1_{INTR} = 38\%$, $F1_{MDDM} = 42\%$, $F1_{CRTX} = 49\%$) (Fig. 4a). This indicates that activation patterns can reliably classify the 12 tasks, even when constrained to a portion of the brain that was active across all tasks. Furthermore, the ability to classify increases as information from the rest of the cortex is included.

**Superior task classification when analysing network connectivity states.** Contemporary network science theory[10] emphasises how cognitive processes are supported, not only by localised brain activity, but also by coordinated interactions that occur across complex coalitions of brain regions. Building on this, we tested whether the functional interactions between brain regions, that occur during cognitive processing, could be used to classify the tasks with a greater accuracy than BA, alone. We applied the same machine learning pipeline using dynamic functional connectivity (dFC), which estimated the change in connectivity between the rest and task blocks, from each ROI pair, and each ROI set. When trained using dFC patterns, the accuracy of the classifiers was significantly higher ($t_{198} = 38.433$, $p < 0.0001$) than for BA for each ROI set (Fig. 4d). Furthermore, the CRTX dFC model, capturing the broadest set of connections in the brain, significantly ($F_{2,297} > 2k$, $p < 0.0001$) outperformed both of the MDDM and INTR dFC models ($F1_{INTR} = 43\%$, $F1_{MDDM} = 43\%$, $F1_{CRTX} = 69\%$). Notably, this difference is unlikely to relate in a trivial manner to the number of features available for classification. For example, the CRTX ROI

connectivity model outperformed the CRTX voxel-wise activity model, despite the latter having more than four times as many features (Table 1). Therefore, in accordance with network sampling theory, coalitions of brain regions, support diverse tasks by transiently adopting distinct connectivity configurations.

**Neuronal activity and connectivity provide complementary information when classifying cognitive states.** The contrast between BA and dFC classification performances raised the question of whether these patterns provide complementary insights into task-evoked network states. To address this question, we trained two-stage stack models, which referenced activation and connectivity together. Specifically, models for voxel-wise activation, and ROI dFC were trained independently at Stage 1, and a stack model was trained using the two resultant sets of 12-way classification metrics at Stage 2. This two-stage approach was designed to account for the difference in the number of features for each type of measure. The stack models outperformed ($p < 0.0001$) the corresponding individual models in all cases ($F1_{INTR} = 52\%$, $F1_{MDDM} = 56\%$, $F1_{CRTX} = 74\%$, Fig. 4g).

**Predicting multi-factorial psychometric structure from task-evoked neuronal states.** We next set out to test a further prediction from network sampling theory, whereby the similarity in the neural states evoked by tasks should relate to their psychometric similarity. Notably, the classification analyses thus far revealed great variability in terms of each model's f1-micro score (an aggregate measure of the classification accuracy of individual tasks) (Fig. 4b, e, h). This indicates that classification accuracy of an individual task depends on its relationship to the rest of the task battery, ROI set (INTR, MDDM or CRTX) and the metric being used for classification (BA,

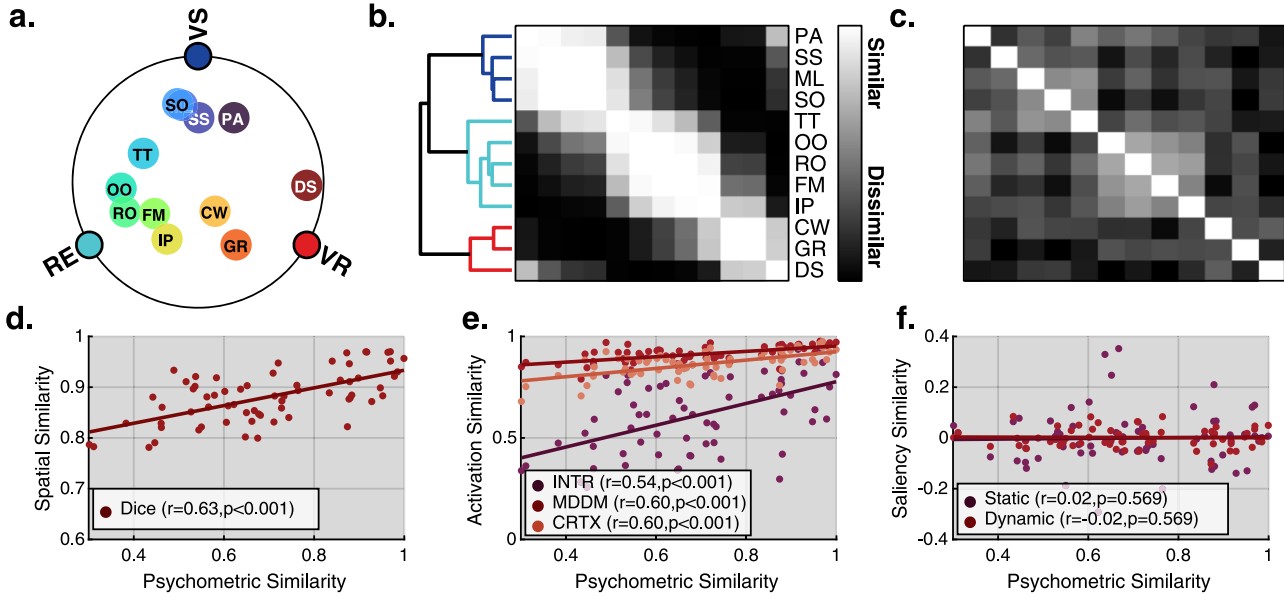

**Fig. 3 Distributed activation pattern correlates with behavioural similarity. a** A previously published internet-based study using test scores from >60k members of the general public was used to define the psychometric similarity between the 12 tasks[2,7]. Loadings from the three-factor solution are presented using a radial projection where each pole represents one of the following factors (VS—visuospatial, RE—reasoning, VR—verbal reasoning). The initials for each task are displayed (see Fig. 1 for reference and Supplementary methods A.1 for a description of each task). **b** Hierarchical cluster analysis of the 66-paired psychometric similarities further reflects the pairwise similarity between tasks. The three-factor structure is clearly evident as clusters of higher correlation proximal to the lead diagonal. **c** DICE coefficients from each pair of whole-brain group-level task activation maps. **d–f** The following scatter plots test association between psychometric similarities and different sources of similarities using a right-tailed Pearson correlation coefficient test. When applicable tests are FDR corrected. **d** A strong positive correlation ($p = 6.82e-09$) was observed between task psychometric similarities and spatial activation DICE coefficients, i.e., tasks that activate the same regions are more proximal in the behavioural factor space. **e** Cross-correlation pairwise similarity extracted from activation patterns within each of the three ROI sets (cortex—CRTX ($p = 1.6e-06$), Multiple-demand default mode—MDDM ($p = 8e-08$), Intersection—INTR ($p = 8e-08$)) showed a strong positive correlation with psychometric similarity (see similarity matrices in Fig. 2f). **f** No significant correlation was observed between task psychometric similarities and similarities in salience time courses (see Supplementary methods A.4). This indicates that the association between brain activation patterns and psychometric similarity is not driven by temporal similarities of stimuli presentation and motor sequencing across the tasks. Source data are provided as a Source Data file.

dFC or stack). To demonstrate this, we quantified binary classification performances between each pair of tasks (66 in total) for each ROI set, and input metric, and correlated these with the corresponding psychometric distances between each task pair. In all ROI sets and metrics, there were moderate negative correlations between binary classification accuracy and psychometric distance between task pairs: BA—$r_{CRTX} = -0.47$, $r_{MDDM} = -0.49$, $r_{INTR} = -0.42$ (Fig. 4c); dFC—$r_{CRTX} = -0.47$, $r_{MDDM} = -0.48$, $r_{INTR} = -0.51$ (Fig. 4f); stack—$r_{CRTX} = -0.50$, $r_{MDDM} = -0.54$, $r_{INTR} = -0.55$ (Fig. 4i). All $p$ values were <0.001 and survived Bonferroni correction ($n = 9$, alpha = 0.05, alpha/$n$ = 0.006). This strongly indicates that psychometrically similar tasks evoke similar functional network states.

**Meta-clusters reflect behaviour complexity.** Behaviour can be viewed as a product of perceptual, cognitive and action systems interacting together[26] and tasks can differ within any combination of these domains. Therefore, we sought to determine which of these aspects of the task designs within our battery contributed to classification of the brain states that they evoke. We clustered the 12 tasks according to psychometric[2,7] (RE, VS, VR), motor interaction type (dynamic, sequence, forced choice), and VS-domain (spatial, digit, object, verbal) dimensions and classified these categories using the same multi-way classification pipeline trained on BA or dFC from the CRTX ROI set. All of the true BA models (Fig. 5g–i) significantly outperformed the null models in each dimension (psychometric $F1_{True} = 69.8\%$, $F1_{Null} = 33.1\%$, motor $F1_{True} = 71.4\%$, $F1_{Null} = 33.2\%$, VS $F1_{True} = 58.2\%$,

$F1_{Null} = 24.8\%$, all $p < 0.01$). Moreover, the dFC models significantly ($p < 0.0001$) outperformed the BA models in each dimension (psychometric $t_{198} = 55.8$, $F1_{dFC > BA} = 83.48 > 67.94\%$; motor $t_{198} = 56.177$, $F1_{dFC > BA} = 84.58\% > 68.8\%$; VS $t_{198} = 55.7$, $F1_{dFC > BA} = 72.4 > 55.4\%$, all $p < 0.0001$). Having established that tasks could be multi-way classified using any of the three dimensions, we next examined the performances of each of the classes. This was important as the meta-class labels produced imbalanced classes (Fig. 5d–f). Specifically, for each dimension, input metric (BA, dFC) and iteration (iter = 100) true performance was compared to 100 permutations where tasks were randomly assigned into classes, while maintaining the true size of each class. Relative to this permutation distribution, the true models significantly outperformed in the psychometric and motor dimensions, but not for the VS dimension: BA—(psychometric $F1_{True > Perm} = 68 > 53.3\%$, $p = 0.001$; motor $F1_{True > Perm} = 68.8\% > 53.8\%$, $p = 0.002$; VS $F1_{True > Perm} = 55.4 > 48.6\%$, $p = 0.061$); dFC—(psychometric $F1_{True > Perm} = 83.5 > 70.7\%$, $p = 0.002$; motor $F1_{True > Perm} = 84.58\% > 71\%$, $p = 0.002$; VS $F1_{True > Perm} = 72.4 > 68.3\%$, $p = 0.161$).

In a further examination of VS differences across tasks, temporal dynamic similarities were calculated between task pairs (i.e. perceptual dynamics of the tasks) using a dynamic saliency model[27] that was run on videos of the tasks being performed (see Supplementary methods A.4). Here, we confirmed that there was also no relationship between the temporal dynamic patterns (i.e. saliency) and behavioural psychometric similarity (Fig. 2f and Supplementary Fig. 3). Together, these results show that classification of tasks relates to a combination of differences in

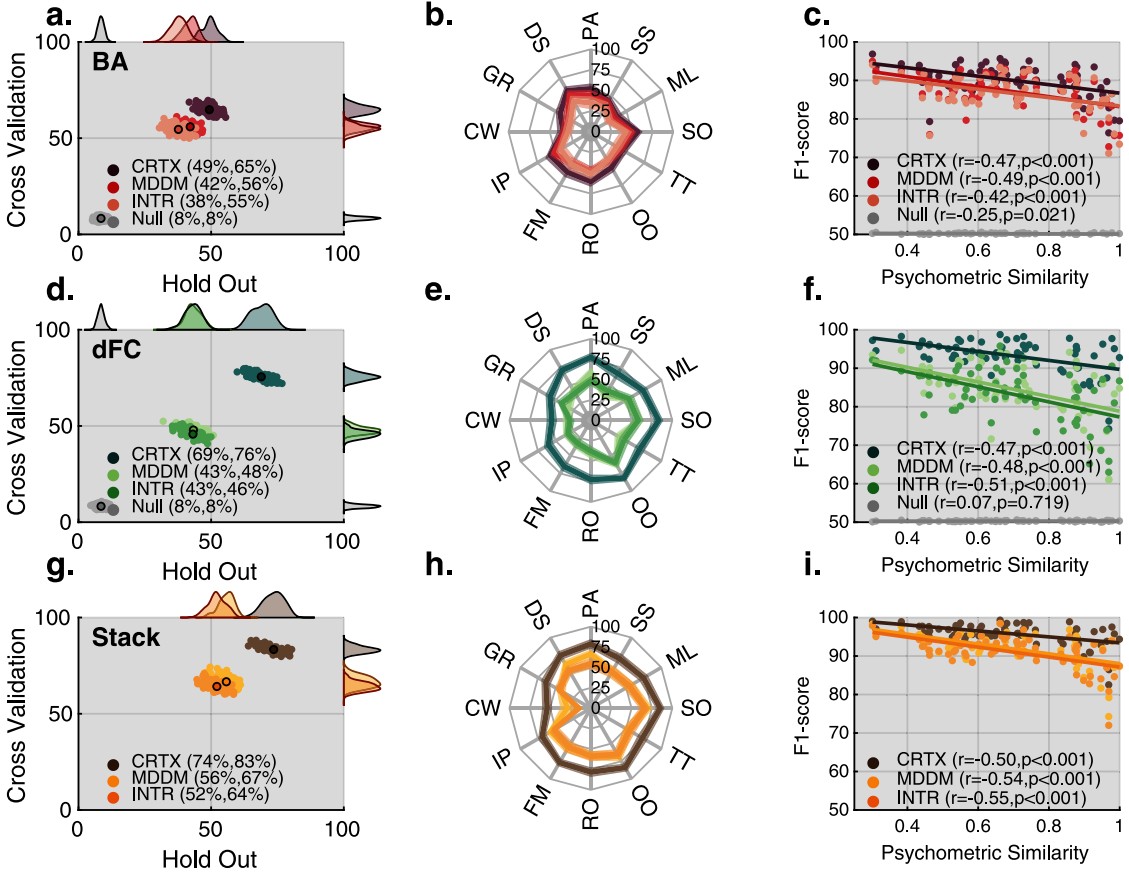

**Fig. 4 Classification analysis across metrics and brain sets.** The 12 tasks can be decoded based on either activation, connectivity or both metrics combined (Stack). The mean pairwise binary classification scores indicate a strong negative correlation with the psychometric similarity implying that behaviourally similar tasks are harder to distinguish. **c**, **f**, **i** The following scatter plots test association between psychometric similarities and different sources of classification accuracies using a left-tailed Pearson correlation coefficient test. All tests are FDR corrected within the comparison. **c** Exact $p$ values: $p_{CRTX} = 0.0001$, $p_{MDDM} = 0.0001$, $p_{INTR} = 0.0003$, $p_{Null} = 0.0204$. **f** Exact $p$ values: $p_{CRTX} = 0.0000$, $p_{MDDM} = 0.0000$, $p_{INTR} = 0.0000$, $p_{Null} = 0.7181$. **i** Exact $p$ values: $p_{CRTX} = 0.1e-04$, $p_{MDDM} = 0.02e-04$, $p_{INTR} = 0.019e-04$. **a**, **d**, **g** Scatter plot of 12-way classification accuracy (F1-score) distributions across ROI sets (i.e. CRTX, MDDM and INTR). Y-axis values were estimated using within sample cross-validation and X-axis represents performance over subject independent held-out test set. **a** Voxel-wise BOLD activation (BA), **d** dynamic functional connectivity (dFC), and **g** stack models that combine information from both BA and dFC. Scatter plots show that: (1) models based on CRTX data outperform other ROI sets; (2) cross-validation performance estimation show that the models are slightly overfitted; (3) combining connectivity and activity in a stack framework is evidence that these measures are complementary for classification. **b**, **e**, **h** Radar plot showing the per task f1-micro accuracy distributions of the held-out test-set. Classification accuracy was highly variable across tasks. In dFC, but not in BA, the overall CRTX f1-micro pattern was visually different from the other ROI sets. **c**, **f**, **i** Strong negative correlations were evident between the psychometric similarity measure (x-axis) and binary classification accuracy (y-axis); therefore, pairs of tasks that were behaviourally similar also evoked similar network states in the brain, which resulted in lower classification accuracy. Source data are provided as a Source Data file.

their cognitive and motoric demands, but less so to differences in their VS-perceptual demands.

**Behavioural performance index correlates with the classifiability of task-evoked network states.** A key element of network sampling theory is that intelligence relates to the brain's capacity to dynamically form network states that are optimal for each specific task. To test this prediction, we calculated the overall classification accuracy of the tasks from the network states of each individual who took part in the study. This was calculated as the number of 1-min task blocks that were correctly classified for the individual, with each individual providing a total of 36 blocks (12 tasks with $3 \times 1$ min replication blocks). The individuals' classification accuracies were correlated with a global composite measure of their task performance: specifically, the first unrotated component from a principle component analysis that was

calculated across the set of 12 task scores. Two participants were identified as outliers at this stage in terms of very low classification accuracy and were removed from the following individual differences analysis (for per participant accuracy see Supplementary Table 5).

Positive correlations were evident for the connectivity and stack models. These were statistically significant for the MDDM and CRTX connectivity and stack models at the FDR corrected threshold, and was strongest for the CRTX connectivity model (dFC: $r_{MDDM} = 0.36$, $p = 0.008$, $r_{CRTX} = 0.39$, $p = 0.005$; Stack: $r_{MDDM} = 0.328$, $p = 0.024$, $r_{CRTX} = 0.27$, $p = 0.033$) (Fig. 6b, c).

**Identifying the most relevant cortical connections for classification.** To gain a better understanding of which network features drove the ability to decode the 12 tasks, we first aimed to identify the connections that contributed the most to the classification

**Table 1 Classification analysis results based on held-out test sample and fivefold cross-validation (CV).**

| Metric | Set | F1 (held-out) | | | F1 (5-fold CV) | | | Features |
|---|---|---|---|---|---|---|---|---|
| | | Mean | Std | CI± | Mean | Std | CI± | |
| BA | CRTX | 49.3 | 2.8 | 48.8–49.9 | 64.8 | 2.1 | 64.4–65.2 | 86,704 |
| | MDDM | 42.1 | 2.5 | 41.6–42.6 | 55.9 | 1.8 | 55.5–56.3 | 31,973 |
| | INTR | 37.7 | 2.9 | 37.1–38.2 | 54.6 | 2.1 | 54.1–55.0 | 14,250 |
| dFC | CRTX | 68.9 | 3.7 | 68.2–69.6 | 75.6 | 1.9 | 75.2–75.9 | 19,900 |
| | MDDM | 43.3 | 3.1 | 42.7–43.9 | 47.9 | 1.8 | 47.5–48.2 | 1326 |
| | INTR | 43.1 | 3 | 42.5–43.7 | 45.76 | 1.95 | 45.4–46.1 | 190 |
| Stack | CRTX | 73.6 | 3.7 | 72.8–74.3 | 83.4 | 1.8 | 83.0–83.7 | 24 |
| | MDDM | 55.7 | 2.8 | 55.2–56.3 | 66.6 | 2.9 | 66.0–67.2 | 24 |
| | INTR | 52.1 | 3.2 | 51.5–52.8 | 64.2 | 2.3 | 63.7–64.6 | 24 |

Classification accuracy measures were approximated using 100 models trained on independent random training samples and accuracy was quantified within training set using fivefold cross-validation and on test dataset sampled from 25% of the participants (see 'Methods' section). Mean F1 measure was calculated as the average F1-macro score across models, reflecting the overall accuracy of the 12-way classification. Std F1 was also calculated, i.e., the standard deviation of the performance distribution and CI represents the 95% confidence intervals. Significance was estimated by calculating the proportion of null models (trained on permuted task labels and it was ~8.3%) that outperformed the mean F1 from the true model, all results have significance p < 0.01.
BA BOLD activity, dFC dynamic functional connectivity, Stack stack model combining BA and dFC, CRTX cortex, INTR intersection, MDDM multiple-demand default mode.

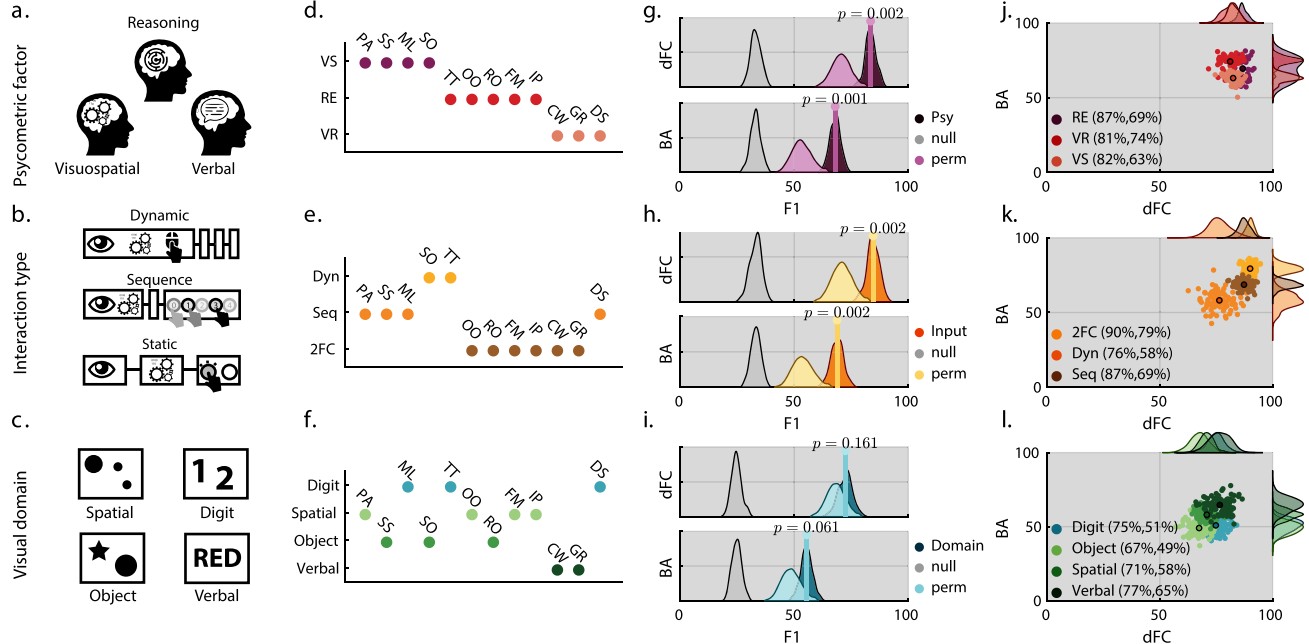

**Fig. 5 Behavioural factors decoded from cortical information.** The design of each task was clustered according to psychometric, motor and perceptual dimensions. **a** Psychometric class defined based on the behavioural factor structure. **b** Motor class. Dynamic interaction tasks require multiple perceptual-mental-action cycles for completion, e.g. the self-ordered search (SO) task. Sequence tasks require the perceptual encoding of a spatial or temporal stimuli sequence coupled with a motor sequence that reflects that encoding. Finally, static tasks require encoding a static stimuli that requires some mental operation culminating in a forced choice response. **c** Perceptual-visual class. Composed of spatial configuration, digit, object and word stimuli. A video of each task is provided in the Supplementary movies 1–12. **d–f** Task to class association showing how each set of classes has a distinct mapping across the tasks. **g–i** Connectivity/dFC (top row) and voxel-wise activation/BA (bottom row) multi-class classification distribution accuracies (F1-score). Grey colour representing null model, which is an approximation of chance based on shuffled response vectors. Dark colour representing true model performance on the held-out test sample and light colour the random permutation of proportional task to class assignments. p values indicate the significance of the distance of the mean true distribution from the permuted distribution using empirical p value approximation. Overall performances are substantially better than randomly clustered tasks. Behavioural-psychometric and motor classes performed better than visual classes. **j–l** Scatter plot of per class f1-minor accuracies. Y-axis represent activation models and X-axis connectivity models, showing that internally the models are not affected by the class imbalance. Source data are provided as a Source Data file.

models. Using a leave-one-out sparse L1 multi-class learning approach, we visually examined the top 0.1% of task-specific and task general connections that were most relevant to the 12-way classification problem. Specifically, 58 sparse multi-way models were trained (one for each participant). The positive, and negative weights from each model were independently binarised and averaged across tasks. The positive weights reflect the connections that transform measures to a positive domain and can be interpreted as task specific. In contrast, the negative weights indicate common connections across the remaining eleven tasks and can be interpreted as task general. We used functional RSNs to visually compare these two complementary aspects of the classification

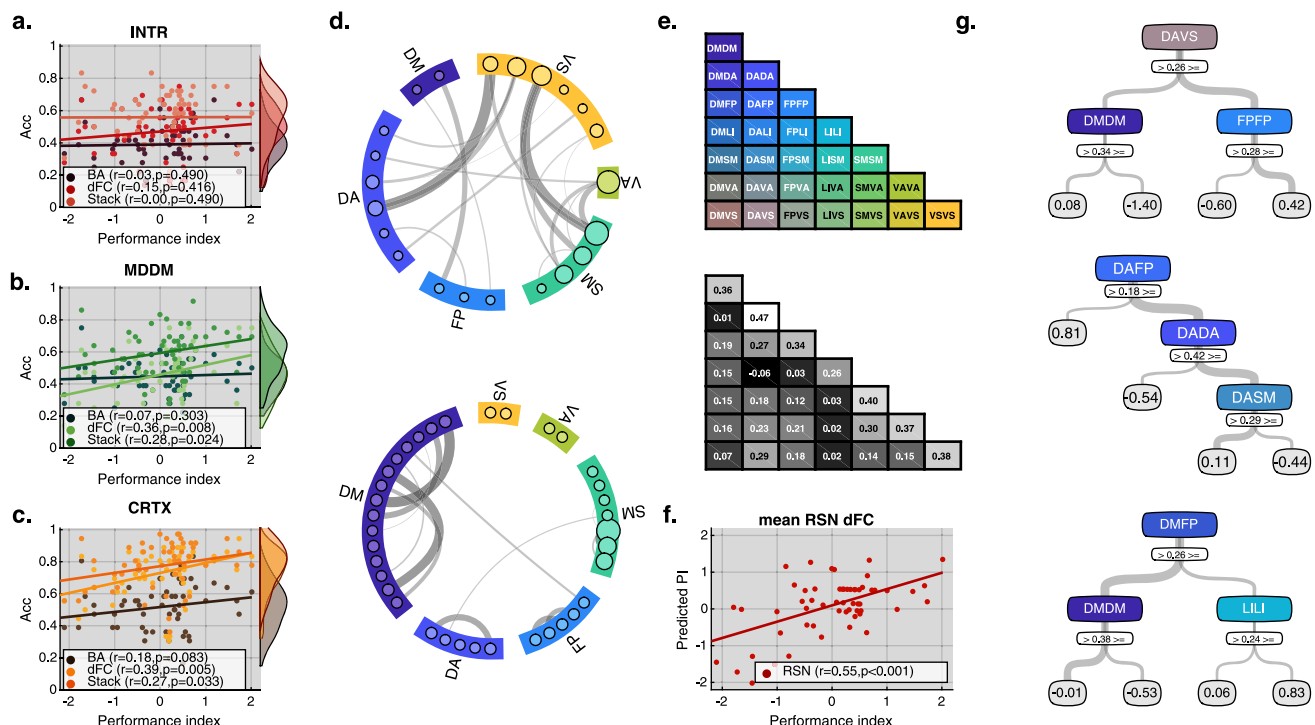

**Fig. 6 Cortical connectivity in networks previously defined by resting state analysis (RSN)[21] predicts individual task performance. a–c** The following scatter plots test association between overall performance index (PI) across all tasks (x-axis) and classification accuracy (y-axis) using a right-tailed Pearson correlation coefficient test. All tests are FDR corrected. Individuals with higher scores express more accurately classifiable task-evoked network states. Best classification was achieved by the stack models. The strongest correspondence to intelligence was produced by the connectivity (dFC) models. **d** Schema-ball plots showing top 0.1% of connections that were stable across 58 (i.e. one for each participant) leave-one-out sparse one-vs-all multi-class models. Top panel shows positive weights used to differentiate any one task from the other 11 and is dominated by connections that are between RSNs. Bottom panel shows the negative weights used to identify features that are similar for most tasks and is dominated by connections that are within RSN. **e** Top panel diagonal shows the seven within RSN labels (i.e. Default mode = DM, Dorsal attention = DA, Frontoparietal = FP, Limbic = LI, Somatomotor = SM, Ventral attention = VA and Visual = VS). The lower triangle shows the 21 unique between RSN combinations. Colours correspond to those in **d** and **g**. Bottom panel shows the mean dFC for each of the RSN combinations averaged across the 12 tasks. Using cross-validated leave-one-out grid search we identified the optimal ensemble boosted structure with respect to the maximal number of branch node splits and the maximum number of trees to train (converged on 3 and 3). We then quantified by 58 independent boosted regression trees. **f** Scatter plot showing the predicted performance index (y-axis) is significantly association ($p = 3.59e{-}6$) with the actual performance index (x-axis) as quantified by 58 independent ensemble trees using a right-tailed Pearson correlation coefficient test. **g** Boosted regression trees trained on the entire sample (i.e. not cross-validated) identified the most important combinations of features for explaining individual differences in performance. Increased connectivity between dorsal attention (DA) and visual (VS) systems was strongly associated with better performance. In contrast, increased connectivity within the default mode (DM) network combined with decreased connectivity between either DA to VS or DM to frontoparietal (FP) was associated with lower performance. Source data are provided as a Source Data file.

models, as they have been previously linked to various intelligence measures[21,28]. The most important task specific connections (top panel Fig. 6d) were between RSNs. The most important task general (bottom panel Fig. 6d) connections were within the RSNs. Furthermore, as expected, task general features were more prevalent (task general: min = 0.54, mean = 0.58, max = 0.67; task-specific: min = 0.37, mean = 0.41, max = 0.5). These results indicate that task-optimised network states involve a perturbation from the resting state architecture of the brain[29].

**Connectivity in resting state networks predicts performance index.** Building on the previous results, a likely possibility is that individual differences in human intelligence performance index relates to non-additive combinations of network features. We simplified the CRTX ROI connectome, from ~20k to 28 features by averaging the within and between RSN connections and used these to train an ensemble of boosted regression trees to examine the non-additive connectivity-performance index relationship (Fig. 6e). Using a minimal set of boosted regression trees, we quantified the

predictive power of this reduced RSN feature set and the most relevant connections for prediction. To avoid over-fitting, we used the same leave-one-out CV approach as before. A cross-validated grid search identified the optimal ensemble boosted structure with respect to the maximal number of branch node splits and the maximum number of trees to train. This converged on $3 \times 3$ tree structure. Positive correlations were evident across the leave-one-out values and cognitive performance index (MSE = 0.6353, $r = 0.55$, $p < 0.001$, Fig. 6f). VS examination of a boosted regression tree for all participants (Fig. 6g) showed that across all trees, the root converged on different between RSNs. However, the majority of branches were composed of within RSN connections. Importantly, in an ensemble of boosted trees, each tree contributes a single multi-step rule, and the predicted value is composed as a sum of leaves across all trees. Furthermore, the width of the branch reflects the generalisation of the rule (concerning the sample), with wider branches affecting more samples. Analysis of the optimal tree combination showed that increased connectivity between dorsal attention (DA) and VS systems strongly associates with better performance. In contrast, increased connectivity within the default

mode (DM) network combined with decreased connectivity between either DA to VS or DM to frontoparietal (FP) associates with lower performance. These results accord with the network sampling view that intelligent behaviour is a function of distributed networks across the brain, task performance involves an optimal perturbation of the network architecture from the steady state, and that certain features within the network have more general and more prominent roles in intelligence test performance.

## Discussion

The results presented here are highly compatible with a network science interpretation of Thomson's sampling theory[14]. Indeed, as has been noted by others, the relationship between the classic notion of a flexible pool of bonds and the analysis of the brain's dynamic networks as applied a century later is striking[17]. Thomson proposed that mental tests tap bonds from a shared pool of neural resources, which is confirmed by our observation that different cognitive tasks tend to recruit unique but heavily overlapping networks of brain regions. Furthermore, when testing Thomson's proposal that the correlation between any two tasks is a function of the extent of overlap between their bonds, we confirmed that the similarities of tasks in multi-factor behavioural psychometric space correlated strongly with the similarities in the dynamic network states that they evoked. These findings corroborate the key tenets of network sampling theory, further predictions of which were tested utilising a combination of machine learning techniques applied to the fMRI and psychometric data.

From a network science perspective, our results showing that the tasks were 12-way classifiable with high accuracy based on their dynamic network states is highly relevant. Indeed, the 74% accuracy achieved by the CRTX stack model was surprising, given that chance was 8.3% and we used just 1 min, comprising 30 images, of task performance data per classified sample. Although activity and connectivity provided complementary information when combined in the stack models, classification accuracy was consistently higher for connectivity when the measures were analysed independently. These results strongly support the hypothesis that the human brain is able to support diverse cognitive tasks because it can rapidly reconfigure its connectivity state in a manner that is optimal for processing their unique computational demands[8,9,12,17]. A key finding was that the task-evoked dynamic network states were consistent across individuals; i.e., our trained 12-way classification models operated with high accuracy when applied in a robust CV pipeline to data from individuals to whom they were completely naive. This was with an out of the box classifier with no CV optimisation, which is important, because it means that the features that drove accurate classification must reflect on a fundamental level how networks in the human brain are prewired to flexibly support diverse tasks.

At a finer grain, these task-optimised network states are most accurately described as a perturbation away from the RSN architecture of the brain[12,29]. More specifically, it was not simply the case that the relative levels of activity or connectivity within each RSN change, i.e., reflecting different mixtures dependent on task demands; instead, the features that were most specific to a given task-evoked state were predominantly the inter-RSN connections. Put another way, task-evoked states are not a simple blending of RSNs, but a dissolution of the RSN structure. This extends the findings of another recent study, where we used a similar analysis pipeline to examine how different aspects of working memory affected brain activity and connectivity[12]. Mirroring the current findings, we found that behaviourally distinct aspects of working memory mapped to distinct but densely overlapping patterns of activity and connectivity within the brain. Taken together, these results do not accord well with the hypothesis that the human brain is organised into discrete static networks. Instead, it would appear that the dynamic network coding mechanism is very high-dimensional, relating to the greater number of possible combinations of nodes[8,9]. There are dependencies whereby some nodes operate together more often than others, but these canonical network states, which are consistently evident in data-driven analyses of the resting state brain, are statistical rather than absolute. Our more holistic interpretation of the relationship between network states and cognitive processes is further supported by the analysis of the classifiability of task clusters when grouped according to their behavioural dimensions. Specifically, when grouped by psychometric, motor or VS characteristics, the clusters were more classifiable than random task groupings in all cases. It was notable though that psychometric and motor characteristics provided a stronger basis for classification. This is interesting, because it pertains to how the most prominent factors of human intelligence differ operationally. For example, it accords well with process overlap theory[17], which proposes that general intelligence relates most closely to processes that are common across many different cognitive tasks.

More generally, the fact that inter-individual differences in the classifiability of the tasks predicted variability in a general measure of behavioural task performance provides further evidence that cognitive faculties relate to the way in which the brain expresses these task-optimal network states. Previous research into the neural basis of human intelligence has typically emphasised the role of flexible FP brain regions[2,30–32]. In this context, our focused analysis of the INTR ROI set warrants further discussion. Brain regions within the INTR ROI set belong to the classical MD cortical volume, which has been closely associated with general intelligence. MD includes the FP brain regions that have the broadest involvement in cognitively demanding tasks[19,20,30]; this includes executive functions, which enable us to perform complex mental operations[33,34] and that have been proposed to relate closely to the 'g' factor[17]. From a graph theoretic perspective, MD ROIs have been reported to have amongst the broadest membership of dynamic networks of any brain regions[35] and it has been shown that inter-individual variability in the flexibility of MD nodes, as measured by the degree of their involvement in different functional networks, correlates positively with individuals' abilities to perform specific tasks, e.g., motor skill learning[36] and working memory[37]. Collectively, these findings highlight a strong relationship between the flexibility of nodes within MD cortex and cognitive ability.

Here, we reconfirmed that MD ROIs were amongst the most consistently active across the 12 tasks. However, we also demonstrated that these ROIs were highly heterogeneous with respect to their activation profiles across those tasks. Furthermore, in many cases they were significantly active for most but not all tasks. This variability in the activation profiles even amongst the most commonly recruited areas of the brain aligns with the idea that MD cortex flexibly codes for diverse tasks in a high-dimensional manner. More critically, the internal activity and connectivity of the INTR ROI set was not strongly predictive of behavioural task performance. Nor did it provide the most accurate basis for classification overall, or correspondence to psychometric structure. Extending to the MDDM set provides an improvement, but it was inclusion of the whole cortex ROI set that provided the best predictor of task and behavioural performance. Furthermore, connections between the core set of INTR regions and the rest of the brain featured prominently in all of the above cases. This finding accords with bonds theory, insofar as that theory pertains to the wide variety of bonds that contribute to diverse behavioural abilities. It also accords particularly well with the core tenet of network science that cognitive processes are emergent properties of interactions that occur across large-scale distributed networks in the brain[10,12].

An intriguing aside pertains to the phenomena of 'factor differentiation'. It was originally noted by Spearman[38] that 'g' explains a greater proportion of variance individuals who perform lower on intelligence tests. This finding has been robustly replicated over the subsequent century[5]. Our results provide a simple explanation for factor differentiation. When individuals of higher intelligence perform different cognitive tasks, the dynamic network states that they evoke are more specific. Therefore, there is less overlap in the neural resources that they recruit to perform the tasks. Given the relationship observed here between network similarity and behavioural-psychometric distance, this would be expected to reduce bivariate correlations in task performances and produce a corresponding reduction in the proportion of variance explained by 'g'.

The boosted ensemble of regression trees provided a simple way to extend the individual differences analysis in order to capture not just mixtures but also interactions between network features when predicting behavioural performance. We observed that increased connectivity between DA and VS systems strongly associated with better performance, whilst increased connectivity within the DMN combined with decreased connectivity between either DA to VS or DM to FP associated with lower performance. This accords well with previous studies that have shown that these networks update their connectivity patterns according to the task context[35,39–43]. However, it was particularly notable that inter-RSN connections again played the most prominent role insofar as they formed the roots of all of the trees, meaning they had the broadest relevance across individuals. This further accords with the view that task-evoked network states are best described as a perturbation from the RSN architecture[12,29].

In summary, we validated multiple key predictions of network sampling theory. This theory can potentially explain key findings from behavioural psychometrics, experimental psychology and functional neuroimaging research within the same overarching network-neuroscience framework, and bridges the classic divide between unitary and multi-factorial models of intelligence. Given that our machine learning analysis pipeline aligns naturally with multivariate network coding whereas more commonly applied univariate methods do not, we believe that the analysis of multivariate network states as applied here has untapped potential in clinical research; e.g., providing functional markers for quantifying the impact of pathologies and interventions on the brain's capacity to flexibly express task optimised network states[11,29].

## Methods
### Behavioural data acquisition
*Ethics approval.* This study was approved by the University of Western Ontario ethics committee and all participants provided written consent before taking part in the study.

*Participants.* Sixty adults (35 females, mean age 22.95, range 18–38 years of age), all with normal hearing and corrected to normal vision were included in the study. Participants were recruited from the University of Western Ontario and surrounding area.

*Intelligence test battery.* All participants engaged with 12 cognitive tasks designed to measure planning, reasoning, attention, and working memory abilities that are believed to be core intelligence abilities. All tasks designs and behavioural scores are reported in detail in the Supplementary methods A.1 and Supplementary movies 1–12. Before scanning, participants underwent a short training session to ensure that they could perform all 12 tasks. The training consisted of reading written instructions followed by one practice block of each task, undertaken on a laptop outside of the MRI scanner. Each participant then undertook 12 functional runs, one for each specific task. These were administered in a predefined order. Each experimental run contained three blocks each 1-min long, separated by 20 s of rest. Tasks were displayed on a projector screen, visible from the bore of the MRI scanner via a mirror. Responses were taken with a custom MRI compatible mouse. In the imaging study, the tests ran as stand-alone software within the Adobe AIR run-time environment.

### MRI acquisition, preprocessing and quality control
*MRI acquisition.* Whole-brain images were collected using a 3 Tesla scanner (TIM Trio, Siemens, Erlangen, Germany). FMRI data were collected across 12 runs (60 min total). During functional scans, a T2-weighted echo-planar image depicting blood oxygenation level-dependent (BOLD) contrast was acquired every 2 s. The first ten images were discarded to account for equilibrium effects. Images consisted of $36 \times 3$ mm slices, with an $80 \times 80$ matrix, $240 \times 240$ mm field of view, TE = 30 ms, flip angle = 90°, echo spacing = 2.65 ms. A 1 mm resolution MPRAGE structural scan was also collected for each participant with a $256 \times 240 \times 384$ matrix, TI = 900 ms, TR = 2.3 s, TE = 2.98 ms and 9° flip angle.

Preprocessing was performed using SPM12 (Statistical Parametric Mapping Welcome Department of Imaging Neuroscience), FSL (FMRIB Software Library v5.0) and MATLAB 2016b (for full description see Fig. S3 and section C of the Supplementary methods B). Specifically, we performed slice timing correction, motion correction, realigned internally, rigidly co-registered to the native structural volume and non-linearly normalised onto MNI space using a DARTEL group template constructed from the structural scans of all individuals. All volumes were then smoothed with an 8 mm³ full width at half maximum Gaussian kernel.

*Imaging quality control.* Signal to noise ratio (SNR) metrics were extracted from the unprocessed fMRI images using an in-house implementation of the metrics proposed by Friedman[44]. An outlier's detection analysis was performed to detect low values (SNR <5). No scans were discarded at this stage.

### Data mining and descriptive analysis
*Group task activation maps estimate.* Using the standard SPM12 mass univariate GLM pipeline for each task first level models were estimated across individuals using the canonical HRF convolved experimental onsets and matrix of nuisance variables (see Supplementary methods C and Supplementary Fig. 7). A second group-level map was generated for each task containing voxels that were consistently activated above baseline across the group. Minimal cluster size was derived by performing uncorrected analysis with a relaxed threshold ($p < 0.01$), then we used the minimal false discovery rate (FDR) cluster ($P_{FDR} < 0.05$) to generate a cluster corrected map for each of the contrasts in the specific conjunction.

*Conjunction analysis.* Conjunction is defined as a logical AND statement between two or more truth conditions. Following Nichols et al.[45] we took the conservative approach of identifying a conjunction volume between two or more tasks simply by intersecting the statistical maps thresholded at a specified alpha rate. We employed an alpha of 0.001 FDR corrected and multiplied the logical mask with the minimum statistical value. This resulted in an intersection volume representing voxels that are active for all 12 tasks.

### Brain masks and region of interest sets.
Pair-wise connectivity between ROIs in the brain is estimated using atlases that aggregate together voxels based on some similarity measure. We use three different ROI sets (see Supplementary Tables 2–4).

*Data-driven ROI intersection set (INTR).* Using the intersection volume created using the conjunction across all 12 task, we generated a data-driven parcellation set using an in-house implementation of the watershed transform[23].

*Multiple-demand default mode ROI set (MDDM).* We used the previously published MD[20] averaged *t*-statistics from contrasts that isolated cognitive demand and averaged across seven different tasks and symmetrised across both hemispheres to define task-based positive and negative BOLD activity and applied the same watershed algorithm to generate data-driven parcellation sets.

*Cortical resting state ROI set (CRTX).* We used an unbiased functional cortical ROI set that clusters 200 independent ROIs on the cortex. It is based on a multi-session hierarchical Bayesian model applied on several large resting state datasets[24] followed by a population-level parcellation of the cerebral cortex into large-scale resting state networks based on similar corticocortical functional connectivity (FC) profiles.

### Behavioural, saliency and BOLD task similarity measures
*Task psychometric pairwise similarity estimate.* How psychometric similar two tasks are can be derived by examining similarities and differences in task performance across a large sample of tests. Here we rely on a three-dimensional behavioural-psychometric model (PCA) previously published[2,7] based on two online datasets ($N = 44{,}780$ and $N = 18{,}455$) using the same cognitive tasks. A VS representation of this psychometric distance is plotted using radial projection where each point represents a task, and each pole represents a factor, the relative distance of a task from each pole reflects its association with that factor. We then calculate the pairwise Euclidean distance (pD) between all possible task pairwise combinations. Using a radial basis function kernel $S(x, x^t) = \exp(-\text{pD}^2)$ we transform the distance measure to similarity measure to simplify interpretation. Finally, we apply hierarchical clustering to this matrix, pairing tasks together based on the relative similarity from each other.

*Task dice coefficients pairwise similarity estimate*. Using the Dice similarity coefficient, also known as the overlap index we measured the overlap between task pairs FDR thresholded voxels to assess pairwise spatial similarity. Dice metric ranges between zero and one, where one indicates complete correspondence between both maps and zero suggests no agreement. Calculated at the voxel level, it is the number of shared voxels divided by the total number of voxels across both volumes. Formally using a confusion matrix, it is defined as $DICE = \frac{2\,TP}{2\,TP+FN+FP}$

where TP (true positives) are the shared voxels in each map. FN (false negative) and FP (false positive) are voxels activated in only one map.

*Task voxel-wise BOLD activation similarity estimate*. Using each of the ROI sets defined above we extracted for all voxels contained within a specific ROI set the sum of beta coefficients per block. This was then averaged across participants and task replications to form a vector of task activation's per task. Similarity was then estimated using Pearson's correlation.

*Task dynamic saliency pairwise similarity estimate*. How similar are the temporal attention dynamics between two tasks is a complex and open question. To approximate this measure we first estimated for each task it's temporal dynamic saliency for 1 min and summed each frame to form a vector that reflects the attention load across time. Our similarity measure is the simple cross-correlation matrix across all possible task pairs.

**Classification analysis**. Classification models have been used in neuroimaging studies for the past two decades[46–48] and can be considered an established method of investigation. However, many pitfalls and biases can be introduced by parameter tweaking to boost reported performance of the learner algorithm. As the objective of this study is not to establish that fMRI data can be used to decode cognitive states, nor introduce a new way to decode such data, but rather use decoding as means of interrogating and comparing between different metrics and volumes of interest we rely on functions from the standard MATLAB Statistics and Machine Learning Toolbox without using any parameter optimisation (see Supplementary methods C).

**Data structures**. In this study we rely on two different BOLD metrics, voxel-wise BA and dFC. Each metric was mined from the three different ROI sets defined above (i.e. CRTX, MDDM and INTR). Data metrics from each participant, across all tasks, and replication blocks were collected for each ROI sets and metrics, and include events from 60 subjects across 12 tasks, with three independent replication blocks for each task. Each block is estimated from 1 min of BOLD time series. As a result, we have $3 \times 2 = 6$ different datasets each containing $60 \times 12 \times 3 = 2160$ matched events with variable feature size.

*Voxel-wise activation*. For each subject across all tasks and for each brain ROI set, we extracted for all voxels contained within a specific set the sum of beta coefficients per block.

*Dynamic functional connectivity*. In neuroscience, FC commonly refers to the statistical association between distal regions across the brain[49]. Estimation of connectivity is commonly performed using some form of dimensionality reduction, either by defining common networks or ROI. Furthermore, it is well established that task performance modulates dFC at different temporal scales, including that of seconds to minutes. For example, Laumman et al. found that performance of external tasks alters short-term FC[50]. Estimation of task-based dFC is commonly conducted using generalised psycho-physiological interactions (PPI)[51]. However, isolating task depended intrinsic connectivity changes using PPI depends on the successful removal of task-induced activity. Recently, it was demonstrated that such removal is better achieved using a finite impulse response (FIR) modelling rather than the canonical HRF model used by the PPI. Therefore, here we followed Cole et al.[52] guidelines and estimate FIR dFC as described in detail in Supplementary methods B.4.

**Classification algorithms**. We used the 'Error Correcting Output Codes' (ECOC) ensemble approach to solve the multi-class problem. The MATLAB built in function with linear support vector machine binary[53] classification was used as the basic learning algorithm to discriminate between one task and all other tasks using either one-vs-all (OVA) or One-vs-one (OVO).

*OVA and OVO multi-class classification schemes*. Importantly, we distinguish between the OVA and OVO classification schemes, as they can be used to answer two different questions. The OVA model builds on identifying patterns that differentiate between one task and all other tasks, in other words domain-specific patterns. While the OVO builds on identifying patterns that differentiate between all pairs of tasks and can be seen as some form of pairwise similarity measure. Both schemes create a multi-dimensional reduced space with task-specific multivariate weighted code.

*Stack classification*. The two-stage stack modelling involves adding an additional layer to the classification processes. Instead of using a coding procedure on the 12-

way positive binary scores (PBS) to assign a unique class to an event, we stacked the three matrices ($12 \times 2$) and trained an additional model on these. To avoid over-fitting, which is common in large *n*, small *m* datasets we used an internal fivefold CV and populated the PBS from the held-out fold. To assess performance, for each metric we used the five CV models, i.e. with completely independent samples to the training set, to estimate PBS for the test set and took the mode (i.e. majority vote) across these scores to get the final test labelling.

*Sparse classification*. In order to identify most relevant connections that took part in the multi-way classification we used the same ECOC multi-way framework but changed the basic learning algorithm from the standard svm to a L1-regularised (i.e. lasso) linear logistic regression classifier.

**Ensemble of boosted least square regression trees**. Ensemble boosted methods combine several sequential models to produce a more accurate predictive performance than utilising a single model where each model builds on its predecessor to create a complementary ensemble. Least square regression trees are a form of nonlinear predictive models where recursive partitioning is used to form a set of decision rules that assign values based on some logical criteria. When used in an ensemble the final predictive value is derived (in our case) as a simple sum across the different trees that create the set.

**Machine learning performance**

*Cross-validation: held-out, K-fold and leave-one-out*. Held-out CV (aka out-of-sample) is a popular method to estimate how machine learning models will generalise to unseen data. The general procedure randomly splits the dataset into a train and test groups with predefined proportion (i.e. 75 and 25%). The model is trained using the training subset and performance is reported using the test set. The k-fold procedure is an extension to the held-out where performance is estimated using all events. In our case (i.e. fivefold), the data are separated into five different training and testing samples and five models are trained using the training sub-sample and tested on the unique fold. Leave-one-out CV is a special case of K-fold, with K equal to the number of participants in the sample. In the permutation held-out CV, a large number of random splitting is created and for each a held-out CV performance is estimated. Performance is then reported using the average accuracy of the distribution across models. The benefit of this latter approach is that it is almost impossible to manipulate (i.e. cherry pick) results and it allows for statistical comparisons between different models (e.g. different dependent variables).

*Classification accuracy estimate*. The balance between precision (i.e. the proportion between correctly classified events and all events classified in class) and recall (i.e. the proportion between correctly classified events and the ground truth) commonly known as F1-score is used as our multi-way accuracy score. F1 is considered a more appropriate measure for multi-class classification problems than percent accuracy. The per class *f*1-minor measure is calculated as the harmonic mean precision and recall (i.e. $f1\text{-minor} = \frac{2rp}{p+r}$), and F1-major (global accuracy measure) is calculated as a simple average of the *f*1-minor.

*Accuracy significance*. To quantify classification significance, we used the conservative repeated permutation CV approach to form paired performance estimations as defined in 'Performance estimation' section. We then estimated the empirical probability as $p = \frac{b+1}{m+1}$, where *b* is the number of events where $F1_{null} > F1_{True}$, i.e. the number of events where the permuted null model outperformed the model tested by real data, and *m* is the number of random sampling pairs (100 bootstraps in our case).

*Mean square error*. To quantify performance index prediction, we used a leave-one out approach to form performance estimations for each subject. Mean square error was used as measure of the average squared difference between the estimated values and the actual value.

**Reporting summary**. Further information on research design is available in the Nature Research Reporting Summary linked to this article.

## Data availability
The raw data that support the findings of this study are committed in the OpenNeuro[54] repository under https://openneuro.org/datasets/ds003148/versions/1.0.1[56]. The processed experimental data that support the findings of this study are committed in figshare[55] under https://figshare.com/articles/dataset/Neuroimaging_evidence_for_a_network_sampling_theory_of_human_intelligence/13237316. Source data are provided with this paper.

## Code availability
The MATLAB code used to create the analysis and figures is available online at https://github.com/esoreq/12_tasks_code.git.

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

## Acknowledgements

Some aspects of the data collected here were supported by the Canada Excellence Research Chairs Program (Grant No. 215063) E.S. is supported by MRC project grant MR/R005370/1 awarded to Robert Leech and Adam Hampshire. R.D. is supported by an EPSRC MRes/PhD scholarship at the Centre for Neurotechnology supervised by Adam Hampshire. I.R.V. is supported by a BBSRC New Investigator Grant (BB/S008314/1). The large online behavioural datasets referenced in the psychometric analyses were collected with support from Roger Highfield of the Science Museum. A.H. is an associate member of the UK Dementia Research Institute.

## Author contributions

A.H. conceived and conducted the experiment(s); E.S. devised, implemented and conducted the analysis of the data as well as developing the data-visualisation framework to create all the figures. All authors contributed to writing the manuscript.

## Competing interests

The authors declare no competing interests.

**Additional information**

