## [Peer Review File · Nature Communications]

Reviewers' comments:

Reviewer #1 (Remarks to the Author):

This is an intriguing article that combines fMRI, machine learning, and behavior in an attempt to examine whether there is a common substrate of human intelligence or whether intelligence emerges from the selection of overlapping configurations of functional networks to support specific task performance. There are clear strengths, including the theoretical importance of the questions addressed, the well-crafted presentation of the work, and the use of out-of-sample analyses.

At the same time, I'm regrettably ambivalent about the work, as the data do not provide compelling support for the high-level claims. This concern stems from fundamental issues of task design, aspects of the analytic approach, as well as the absence of critical supporting statistical tests.

Major comments

1. With respect to task design, first it is worth noting that the descriptions of the 12 tasks are insufficient, as much detail is missing and this appears critical for understanding what drives the classifiers' performance. Second, and relatedly, the authors, based on prior literature, carve voxel-space into MD and MDc (along with WC) subsets, and then assume that classification based on MD voxels (and ROIs and networks) must reflect working memory and executive/control processes that are thought to be key substrates of intelligence. However, it is unclear that this assumption is justified. Even based on the very brief descriptions of the tasks, it is clear that there are many low-level perceptual and motor differences between the tasks, along with differences in the temporal profile of when higher-level processes would be engaged within each task. As such, there is no basis to conclude that classification based on MD voxels/ROIs/networks reflects anything other than decoding based on low-level perceptual and motor differences (and/or differences in the temporal profile of shared higher-level processes); there are likely so many uncontrolled differences between tasks that it is simply difficult to rule this out. The fact that classification levels were high for both MD and MDc trained classifiers further motivates this concern (along with prior observations of content-based decoding via frontal voxels and ROIs).

2. With respect to the analytic approach, given the authors' focus on MD vs MDc voxels and ROIs, the decision to include parcels with 35% overlap of the Fedorenko-defined MD is concerning, as this leaves open the possibility that many (65%) of the voxels in these parcels may be more appropriately classed as MDc. A re-analysis that excludes non-overlapping voxels seems necessary, to the extent that the authors feel it is critical to leverage the notion of a MD network in their analysis.

3. With respect to statistical support for the conclusions, I'm concerned that there are multiple instances where quantitative differences in classifier performance or the strengths of correlations are described and used to imply meaningful differences in outcomes, but without the necessary supporting statistical tests to demonstrate that the quantitative differences (e.g., between MD vs. MDc) are significant. I believe there are at least four statements about differences in the strength of effects that are not supported by statistics.

Minor comment

While the Discussion is thoughtful and quite provocative, much of it is speculative. This might be fine in the absence of the above flagged concerns, but it reads as extending well beyond the data (even with the appropriate signaling of which statements are and are not tentative).

Reviewer #2 (Remarks to the Author):

This is a review for the manuscript "Network Sampling Theory of Human Intelligence" submitted to Nature Communications.

The manuscript outlined a compelling framework for human intelligence, “network sampling theory”, which interpreted Thomson’s sampling theory of intelligence in terms of a so-called “high dimensional dynamic network mechanism”.

The authors performed a creative and original set of analyses linking key ideas from the psychometrics of human intelligence to brain function. They successfully performed 12-way classification of the cognitive tasks based on the pattern of BOLD activation as well as functional connectivity during each task. Intriguingly, classifier accuracy was greater for individuals with higher intelligence scores. They also related the psychometric distance between pairs of intelligence tests to the ability of a classifier to discriminate between the brain activation patterns evoked by each pair of tests. Tests with greater psychometric distance exhibited more distinct brain activation patterns. These constitute novel and exciting findings in the field of cognitive neuroscience, grounding observations from psychometrics to characteristics of brain function.

The authors also aimed to evaluate the role of multiple demand (MD) cortex and frontoparietal brain regions in their network sampling theory of intelligence. They showed that a classifier can discriminate between tasks based only on brain activity in MD cortex, and that the accuracy of this classification reflected the psychometric dissimilarity between the tasks. The authors also present activation brainmaps for each of the 12 tasks. The authors conclude that their overlap “accord[s] poorly with the notion of core brain regions that are involved in all cognitive operations... [rather, these regions] are involved in many cognitive tasks”. However, it is not obvious (i.e. quantified) how the data led to this conclusion. In particular, it is statistically inappropriate to make such a statement (i.e. that the null hypothesis is true) based on t-tests. Moreover, this conclusion seems unwarranted given the emphasis on the role of networks (as opposed to univariate activation) in human intelligence.

Overall, the study presents interesting and original results and refines ideas regarding the neural underpinnings of human intelligence, though some issues with methods and interpretation substantially reduce enthusiasm for this study.

Major comments

- Interpretation of the results was muddled by the fact that the sensorimotor demands are not controlled (or, alternately, sufficiently diverse) across the tasks, which all require varying degrees of visual and motor processing. This is demonstrated by the intersection of the activation maps across the tasks, which includes large swaths of visual and motor cortex. This is a substantial flaw in the design of the study, and hinders both its interpretability and generalizability (given the interest in intelligence/cognition rather than sensorimotor processes). This should be discussed as a limitation of the study and its implications explored (e.g. could this be driving classification differences?).
- The authors conclude that their overlap “accord[s] poorly with the notion of core brain regions that are involved in all cognitive operations... [rather, these regions] are involved in many cognitive tasks”. However, it is not obvious how the data led to this conclusion. The overlap of the 27% of activated voxels with MD cortex is not quantified. In particular, it is statistically inappropriate to make such a statement (i.e. that the null hypothesis is true) based on binarized activation maps from many t-tests. Indeed, this is a common logical error in neuroimaging analyses (“imager’s fallacy”), in which lack of overlap in two activation maps is interpreted as a statistically significant difference between maps (see [Henson, R. (2005). “What Can Functional Neuroimaging Tell the Experimental Psychologist?” *The Quarterly Journal of Experimental Psychology Section A* 58 (2): 193–233.]) (This is even more problematic given the very minimal amount of data per task per subject [1 minute], which undoubtedly increases the width of confidence intervals.) Moreover, this conclusion seems unwarranted given the emphasis on the role of networks (as opposed to univariate activation) in human intelligence. Indeed, the ability to decode so many of the tasks based on subtle differences in MD voxel activity seems to support the role of MD regions in general intelligence.
- The key tenants of a “network sampling theory of human intelligence” could be more clearly laid out and contrasted with other theories of intelligence (maybe in a special panel/table/figure).
- I could not follow section 7 and Figure 4 of the results. What is being shown in Figure 4 A? The

reasoning for trying to find the top 3 most-distinguishable tasks is not clear. Why 3? There is no comparison with other iterations of the 3-way classification shown, so the stability of these results is difficult to evaluate. Why not use the 3 assignments from (VS, Re, and Ve) for 3-way classification? Couldn't these results be driven by differences across tasks in stimulus features (e.g. visual and motor) introduced by the experimenters?

- Data from an auditory-imagery task is referred to in section 1 of the results and in figure 1B. The source of this data is not clear and needs to be made explicit. More problematically, the inclusion of these results is objectionable, given that they appear to be from a distinct dataset from the present study. Are these studies well matched to the other task conditions? Why include these results and not others? Does this truly add anything to the manuscript? The claim that MD regions are not involved in "all" cognitive operations appears to rely quite heavily on inclusion of the auditory-imagery task, yet it appears inappropriate to include this result in the present study.
- A recent paper [Siegel JS, Mitra A, Laumann TO, Seitzman BA, Raichle M, Corbetta M, Snyder AZ (2016) "Data Quality Influences Observed Links Between Functional Connectivity and Behavior". *Cereb Cortex*. 1–11. <http://doi.org/10.1093/cercor/bhw253>] established that in-scanner motion is strongly anticorrelated with individual differences in general intelligence, and that this produces spurious correlations with fMRI functional connectivity measures. This puts a large burden on all neuroimaging studies to firmly establish that their effects are not driven by motion artifacts. The present study states in the supplemental materials: "Nuisance experimental matrices were formed using motion parameter estimates (in an extended 24-parameter model) and motion outlier (spikes) events." First, more details are needed to evaluate how likely these steps were to remove the effects of motion on the connectivity (and activity) estimates (e.g., the motion outlier threshold). Second, this is likely insufficient for removing motion/physiological artifacts that correlate with intelligence, as explained in this paper: [Power, Jonathan D., Mark Plitt, Stephen J. Gotts, Prantik Kundu, Valerie Voon, Peter A. Bandettini, and Alex Martin. (2018). "Ridding fMRI Data of Motion-Related Influences: Removal of Signals with Distinct Spatial and Physical Bases in Multiecho Data." *Proceedings of the National Academy of Sciences* 115 (9): E2105–14. [doi:10.1073/pnas.1720985115](https://doi.org/10.1073/pnas.1720985115)]. In particular, it would be important to regress out global signal (or use aCompCor) to reduce the effects of physiological artifacts on the results.
- A series of papers have established that FIR regression (rather than canonical HRF, as is standard in PPI/gPPI analysis) should be used to remove the inflation effects of task activations on task functional connectivity estimates: [Cole MW, Ito T, Schultz D, Mill R, Chen R, Cocuzza C (2019) "Task activations produce spurious but systematic inflation of task functional connectivity estimates". *NeuroImage*. 189:1–18. <http://doi.org/10.1016/j.neuroimage.2018.12.054>] and also [Al-Aidroos N, Said CP, Turk-Browne NB (2012) "Top-down attention switches coupling between low-level and high-level areas of human visual cortex.". *Proceedings of the National Academy of Sciences*. 109:14675–14680. <http://doi.org/10.1073/pnas.1202095109>]. Without using FIR regression to remove the effects of task activations the task functional connectivity analyses are likely biased by the task activations, especially given that classification is used (which can pick up on subtle biases in the data). In particular, the task functional connectivity results cannot be trusted to be specific to connectivity, such that the task activation results might be re-described as "connectivity".

Minor comments

- The brain maps are small and of low quality, especially for figure 4 (where the orange and red are difficult to differentiate). It would be better to plot the brain maps with a higher-quality surface rendering to allow for better visualization of activity patterns.
- It's unclear what Figure 1.C adds to the manuscript. It seems more beneficial to enlarge the other parts of the figure.
-
- It is problematic that there is no Methods section in the manuscript. Many details are unclear despite some brief description of methods in the Results section. A detailed Methods section should be added to the main manuscript (not the supplement). Even in the supplemental methods there is not enough detail. In particular, many of the preprocessing parameters are not included, making it difficult to evaluate the quality of the analyses.
- A more thorough discussion of MD cortex would be helpful for readers. The MD voxel map should be visualized in the main text and the ROI assignments shown in the supplement. (The brain parcellation in the supplement for WC could show the ROI mappings of the brain atlas used.)

- Aspects of the manuscript feel unfocused, not fully fleshed out, and should be made more accessible. The last ~half of the discussion is especially confusing.
- The observation that the so-called WM sub-systems of section 7 overlap “most densely” in MD cortex is not quantified (and/or no reference is provided). It is unclear if this is referring to the data-driven results in the present study or the results from previously published work.
- The correlations presented in figure 2 should probably be corrected for multiple comparisons.
- Was the same null used for the classifications based on ROI, voxel, dFC, and stack? (Only one is shown in figure 1 D.)
- The authors indicated that 12 tasks comprised of 3 blocks of 1 minute duration (separated by 20 s of rest) were collected during BOLD fMRI scans with a TR of 2 s. In this case, it is perplexing how the authors examined 2160 events (60 time points x 12 tasks x 3 blocks), rather than 1080 events (30 time points x 12 tasks x 3 blocks).
- The author’s don’t explore the role of differences in how the tasks load onto ‘g’.
- There could be some nuance in distinguishing between the features of the brain that enable intelligent behavior at all and the features associated with high levels of intelligence.
- It would be nice to comment on how this ties into the “efficiency” framework of intelligence (Schultz and Cole, 2016 [Schultz, Douglas H., and Michael W. Cole. (2016). “Higher Intelligence Is Associated with Less Task-Related Brain Network Reconfiguration.” *The Journal of Neuroscience* 36 (33): 8551–61. doi:10.1523/JNEUROSCI.0358-16.2016.]), which shows results essentially opposite of those presented here.
- Work on the figure legend descriptions. Make sure all axes labeled and legends (or descriptions in figure caption) are present in each figure. Define grey points (presumably null model) on Figure 3C. Work to clarify what each figure is showing, e.g. what points in scatter plots are (pairs of tasks vs individuals), abbreviations, etc. The text at the end of Figure 4 caption seems like it belongs in the main text and the figure is missing a legend. Could say “classifier accuracy (F1-score)” in figures to ease accessibility.
- Edit more thoroughly. Semicolons are overused. There are lots of run-on sentences. There are typos and formatting issues throughout (especially in the supplement).

Response to the reviewers

We thank the reviewers for their critical assessment of our work. Overall the comments were both positive and constructive, highlighting the importance of the research questions, the sophistication of our methods for addressing them and obvious overall strengths of the work. Both reviewers provided particularly insightful, detailed and helpful comments. We have thoroughly addressed all of them in our modified manuscript. In fact, given the sophistication of the data-driven analysis methods applied, it has taken many months of further work to achieve this. We note that one of the reviewers asked us to change the fundamental way in which we estimate functional connectivity to an alternative method.

We applied the suggested alternative approach (i.e., using finite impulse response functions and adding tissue regressors). This demanded the complete rerunning of all stages of our analysis down to the level of single-subject models. Reassuringly, our key results remain essentially the same with either approach. More importantly, implementing the other suggestions enabled us to derive a wealth of further novel insights, and we are very grateful to the reviewers for those insightful suggestions. Also, as requested, we have included substantial new figures to make these results readily accessible.

Our revised article has been dramatically strengthened by addressing these comments. Here we include a detailed point by point response to the reviewers. We hope that you will both share our enthusiasm and consider this substantially amended article suitable for publication in Nature Communications. We look forward to your response.

Reviewer 1

This is an intriguing article that combines fMRI, machine learning, and behavior in an attempt to examine whether there is a common substrate of human intelligence or whether intelligence emerges from the selection of overlapping configurations of functional networks to support specific task performance. There are clear strengths, including the theoretical importance of the questions addressed, the well-crafted presentation of the work, and the use of out-of-sample analyses.

At the same time, I'm regrettably ambivalent about the work, as the data do not provide compelling support for the high-level claims. This concern stems from fundamental issues of task design, aspects of the analytic approach, as well as the absence of critical supporting statistical tests.

Reply: We thank the reviewer for this analytical view of our original manuscript. We believe that by addressing all of the reviewer's points, see below, we have made a strong case to support our findings in a clearer manner.

Major

Reviewer Point P 1.1 —” *With respect to task design, first it is worth noting that the descriptions of the 12 tasks are insufficient, as much detail is missing and this appears critical for understanding what drives the classifiers' performance.* ”

Reply: We thank the reviewer for raising this important point and agree that a clearer description of the tasks is necessary. To address this we have added a new supplementary figure of task design. We expanded the description of the tasks in the supplementary materials and added 12 supplementary one minute movies (one for each task) showing the actual task mechanics in terms of presented stimuli and motor response. For each task movie we also applied a gold-standard dynamic saliency model to quantify and compare the temporal saliency and motor loads of the different tasks based on the dynamic presentation of the stimuli as well as motor responses captured by mouse movement.

Reviewer Point P 1.2 —” *Second, and relatedly, the authors, based on prior literature, carve voxel-space into MD and MDc (along with WC) subsets, and then assume that classification based on MD voxels (and*

ROIs and networks) must reflect working memory and executive/control processes that are thought to be key substrates of intelligence. ”

Reply: In the previous analysis we used a cortical functional parcellation and created two subsets under the assumption that MD (or at least it's thresholded intersection with the cortical parcellation) reflected areas that are active for most cognitive tasks and thus may be viewed as "domain general" areas. Following the same logic we defined MD^c as mostly "domain specific" areas. However, following this comment we changed our analytic approach. We now take an additive rather than comparative approach. Our results show that in "domain general" only areas (this is referred to as the INTR mask), composed of visual and attention systems, classification of tasks is significantly different from chance. Surprisingly, using the original "multiple demands cortex"[?] positive and negative activation areas (i.e. extending both the number of voxels and brain region coverage) did not show substantial improvement compared to the INTR mask. Furthermore, adding information from the entire cortex (i.e. "domain general" AND "domain specific" referred to as the CRTX mask) local BOLD activity information, at the voxel level only improves our classification by 11%. In contrast, while classification accuracy using functional connectivity (dFC) information from the INTR mask is slightly more accurate compared to the voxel level accuracy (i.e. 43% vs. 38%), scaling up to CRTX dFC improves accuracy by 26% with accuracy of 69%. Importantly, as noted by both reviewers this measure is mean out-of-sample accuracy, based on 100 matching independent random sampling separating the data set based on subjects with a 75/25% split.

Reviewer Point P 1.3 —” *However, it is unclear that this assumption is justified. Even based on the very brief descriptions of the tasks, it is clear that there are many low-level perceptual and motor differences between the tasks, along with differences in the temporal profile of when higher-level processes would be engaged within each task. As such, there is no basis to conclude that classification based on MD voxels/ROIs/networks reflects anything other than decoding based on low-level perceptual and motor differences (and/or differences in the temporal profile of shared higher-level processes); there are likely so many uncontrolled differences between tasks that it is simply difficult to rule this out. The fact that classification levels were high for both MD and MD^c trained classifiers further motivates this concern (along with prior observations of content-based decoding via frontal voxels and ROIs). ”*

Reply: With respect to the comment that our classification accuracy is driven by low-level perceptual and motor differences between the tasks, we added a completely new analysis and result section addressing this point specifically. To clarify our view, on a conceptual level we expect, based on network sampling theory, that performance of any task will relate to a unique mixture of brain functions, spanning cognitive, perceptual, decision and motor abilities. We agree that task similarities (and differences) exist in all of these between the tasks, although we did design them to be as similar as possible, e.g., within the same general template and framework. Trying to control these differences is nearly impossible across tasks, as the reviewer correctly noted. If the tasks are visually and motorically all the same, then any inference we could draw would be rather narrow in cognitive scope. The alternative approach to matching all such dimensions, is to quantify and model them. That is what we do in the additional analyses. Specifically we cluster the tasks according to each such factor, and then determine which of them has a validity that is greater than clustering by chance. Surprisingly, our analysis showed that motor and psychometric factors aggregated the tasks better than chance, however, perceptual differences do not. We also discuss in the main text and clarify that it is the common relationships between tasks in network and psychometric space that is critical for network sampling theory, that is regardless of the degree to which such relationships have a basis in visual, motor or cognitive networks in the brain. We feel that these analyses greatly enhance the paper and we are grateful to the reviewer for this.

Reply: With respect to the comment that our classification accuracy is driven by the temporal profile of when higher-level processes would be engaged within each task. Here we also added a completely new analysis and an additional paragraph in the results addressing this point specifically. We approximated the temporal profile of each task using a computational dynamic saliency model, and cross correlated tasks profiles to form a similarity matrix. We then show that while the psychometric factor analysis strongly correlates with volume overlaps between tasks and BOLD activity patterns it has no relationship to approximated temporal profile. Again, this is an excellent

point on the part of the reviewer and we feel that addressing it in this manner greatly strengthens the inferences that we can draw.

Reviewer Point P 1.4 —” *With respect to the analytic approach, given the authors’ focus on MD vs MDc voxels and ROIs, the decision to include parcels with 35% overlap of the Fedorenko-defined MD is concerning, as this leaves open the possibility that many (65%) of the voxels in these parcels may be more appropriately classed as MDc. A re-analysis that excludes non-overlapping voxels seems necessary, to the extent that the authors feel it is critical to leverage the notion of a MD network in their analysis.* ”

Reply: We agree with the reviewer, in fact, intersecting a mask volume onto a surface generated parcellation, was in retrospect, a fundamental analytical mistake. We have now rerun all of our analyses, comparing the whole brain cortical atlas that we originally used in the parcellation map derived from the original Fedorenko analysis, as well as the data-driven intersection map across tasks, both parcellated using a data driven approach. This new analysis takes an additive approach (rather than the previous hierarchical comparative one) where we identify domain-general regions in the strictest data-driven manner (i.e. the intersection). Using the unbiased original statistical volume defining MD (i.e. Fedorenko volume) we define a task positive to negative ROI set. Finally, we use the original cortical resting state parcellation by Schaefer as a feature space containing both domain specific and domain general regions. We also simplified the results by comparing between voxel-wise activations within different ROI sets and the task-based connectivity across these ROI pairs. As expected this slightly changed our reported results, in our opinion in a way that supports our previous assumptions even better. More specifically, we stated before that activation cortical patterns slightly outperformed both constrained subsets ($F1_{CRTX} = 59.3\% > F1_{MD} = 51.3\%, F1_{MDc} = 54.61$). Our new analyses show that these differences are much stronger when comparing domain general areas in a stricter sense ($F1_{CRTX} = 49.3\% > F1_{MDDM} = 42.1\%, F1_{INTR} = 37.7\%$). Notably these results are based on the more strict out-of-sample accuracy measures with the standard cross-validation results showing less extreme differences ($F1_{CRTX} = 64.8\% > F1_{MDDM} = 55.9\%, F1_{INTR} = 54.6\%$). These effects are much more pronounced in the connectivity measures, where we stated before the following results: $F1_{CRTX} = 72.1\% > F1_{MD} = 57.67\%, F1_{MDc} = 68.74$. Our new analyses show a stronger effect for both out-of-sample: $F1_{CRTX} = 68.9\% \gg F1_{MDDM} = 43.3\%, F1_{INTR} = 43.1\%$ and cross-validation: $F1_{CRTX} = 75.6\% > F1_{MDDM} = 47.9\%, F1_{INTR} = 45.76\%$. Importantly, the complementary relationship between BOLD activity (BA) and dynamic functional connectivity (dFC) we previously reported were also preserved. Our previous submission showed that Stack models outperformed both metrics across all subsets ($F1_{CRTX} = 78.46\% > F1_{MD} = 70.66\%, F1_{MDc} = 75.32$). The same relationship also holds in the new analyses for both out-of-sample: $F1_{CRTX} = 73.6\% \gg F1_{MDDM} = 55.7\%, F1_{INTR} = 52.1\%$ and cross-validation: $F1_{CRTX} = 83.4\% > F1_{MDDM} = 66.6\%, F1_{INTR} = 64.2\%$.

Reviewer Point P 1.5 —” *With respect to statistical support for the conclusions, I’m concerns that there are multiple instances where quantitative differences in classifier performance or the strengths of correlations are described and used to imply meaningful differences in outcomes, but without the necessary supporting statistical tests to demonstrate that the quantitative differences (e.g., between MD vs. MDc) are significant. I believe there are at least four statements about differences in the strength of effects that are not supported by statistics.* ”

Reply: We thank the reviewer for raising this point. We have now added fully documented code for each of our figures. We also added both statistical measures and significance values where needed.

Minor

Reviewer Point P 1.6 —” *While the Discussion is thoughtful and quite provocative, much of it is speculative. This might be fine in the absence of the above flagged concerns, but it reads as extending well beyond the data (even with the appropriate signalling of which statements are and are not tentative).* ”

Reply: We thank the reviewer for raising this point. Our new analyses greatly strengthen our previous statements and we have reworked to reduce speculation (although consider some speculation to be valuable in guiding future work). More specifically, our main claims are that:

- Cognitive tasks evoke distinct generalisable configurations of activity and connectivity in the brain.
- The similarity between cognitive task pairs maps to the extent of 'overlap' between recruited neural resources.
- Individual functional dynamic repertoires positively correlates with task performance index, with the top-performers expressing task-configurations that are more reliably classifiable.

Reviewer 2

The manuscript outlined a compelling framework for human intelligence, “network sampling theory”, which interpreted Thomson’s sampling theory of intelligence in terms of a so-called “high dimensional dynamic network mechanism”.

The authors performed a creative and original set of analyses linking key ideas from the psychometrics of human intelligence to brain function. They successfully performed 12-way classification of the cognitive tasks based on the pattern of BOLD activation as well as functional connectivity during each task. Intriguingly, classifier accuracy was greater for individuals with higher intelligence scores. They also related the psychometric distance between pairs of intelligence tests to the ability of a classifier to discriminate between the brain activation patterns evoked by each pair of tests. Tests with greater psychometric distance exhibited more distinct brain activation patterns. These constitute novel and exciting findings in the field of cognitive neuroscience, grounding observations from psychometrics to characteristics of brain function. The authors also aimed to evaluate the role of multiple demand (MD) cortex and frontoparietal brain regions in their network sampling theory of intelligence. They showed that a classifier can discriminate between tasks based only on brain activity in MD cortex, and that the accuracy of this classification reflected the psychometric dissimilarity between the tasks. The authors also present activation brainmaps for each of the 12 tasks.

The authors conclude that their overlap “accord[s] poorly with the notion of core brain regions that are involved in all cognitive operations... [rather, these regions] are involved in many cognitive tasks”. However, it is not obvious (i.e. quantified) how the data led to this conclusion. In particular, it is statistically inappropriate to make such a statement (i.e. that the null hypothesis is true) based on t-tests. Moreover, this conclusion seems unwarranted given the emphasis on the role of networks (as opposed to univariate activation) in human intelligence.

Overall, the study presents interesting and original results and refines ideas regarding the neural underpinnings of human intelligence, though some issues with methods and interpretation substantially reduce enthusiasm for this study.

Reply: We thank the reviewer for their careful reading of our manuscript. We have performed substantial work to improve methodological aspects and we believe that this provided a stronger support for the interpretation of our findings. A point-by-point response is provided below.

Major

Reviewer Point P 2.1 —” *Interpretation of the results was muddled by the fact that the sensorimotor demands are not controlled (or, alternately, sufficiently diverse) across the tasks, which all require varying degrees of visual and motor processing. This is demonstrated by the intersection of the activation maps across the tasks, which includes large swaths of visual and motor cortex. This is a substantial flaw in the design of the study, and hinders both its interpretability and generalizability (given the interest in intelligence/cognition*

rather than sensorimotor processes). This should be discussed as a limitation of the study and its implications explored (e.g. could this be driving classification differences?). ”

Reply: We thank the reviewer for this comment, indeed the fact that all of our cognitive tasks require substantial visual processing and similar motor response (all mouse clicks, albeit with different temporal and spatial patterns governed by task design) is challenging. To examine whether task mechanics (i.e. the difference in stimuli, response type and tempo, etc.) are driving task classification differences we conducted a number of additional analyses. Particularly, we approximated the saliency load across time for each task and compared the pairwise similarity matrix to both the neuronal and psychometric ones. Our results showed no relationship between the approximated temporal saliency profile to any of the group measures (i.e. spatial DICE overlap task similarity, constraint BOLD voxel-wise activation pattern or psychometric factor similarity). Another way to test this is to cluster tasks to meta-classes based on the different conceptual dimension and see how manual meta-classes compare to randomly allocated ones. Our meta-classes classification analysis showed that clustering tasks based on either psychometric factors or motoric response type were significantly better than randomly assigned classes. In contrast, stimuli type clustering was not significantly better than randomly allocated classes. We also discuss this in the main text and clarify that it is the common relationships between tasks in network and psychometric space that is critical for network sampling theory, that is regardless of the degree to which such relationships have a basis in visual, motor or cognitive networks in the brain. We feel that these analyses greatly enhance the paper and we are grateful to the reviewer for this.

Reviewer Point P 2.2 —” *The authors conclude that their overlap “accord[s] poorly with the notion of core brain regions that are involved in all cognitive operations... [rather, these regions] are involved in many cognitive tasks”. However, it is not obvious how the data led to this conclusion. The overlap of the 27% of activated voxels with MD cortex is not quantified. In particular, it is statistically inappropriate to make such a statement (i.e. that the null hypothesis is true) based on binarized activation maps from many t-tests. Indeed, this is a common logical error in neuroimaging analyses (“imager’s fallacy”), in which lack of overlap in two activation maps is interpreted as a statistically significant difference between maps (see [Henson, R. (2005). “What Can Functional Neuroimaging Tell the Experimental Psychologist?” The Quarterly Journal of Experimental Psychology Section A 58 (2): 193–233.]) (This is even more problematic given the very minimal amount of data per task per subject [1 minute], which undoubtedly increases the width of confidence intervals.) Moreover, this conclusion seems unwarranted given the emphasis on the role of networks (as opposed to univariate activation) in human intelligence. Indeed, the ability to decode so many of the tasks based on subtle differences in MD voxel activity seems to support the role of MD regions in general intelligence.* ”

Reply: We thank the reviewer for this comment, we understood from the comment that several things were unclear. First regarding the amount of data we have. Our dataset is composed by 3 replication blocks from each task, each 1 minute long, with interleaved rest. This makes 12 (tasks) * 3 (blocks) * 60 (participants) observations, i.e., 2160 in total for our analysis, as we classify each 1 minute block.

Reply: Secondly, regarding the broader question of what MD is and what role it has in supporting different intelligent behaviours. To clarify our viewpoint, MD could be conceived as one functional network, a set of connected functional anatomical modules, or as a property of the dynamic functional connectome. We hold the latter view and believe that given a set of different cognitive tasks, the functional relationship of any brain region exists on a domain-general domain-specific continuum. From this perspective, the brain regions composing MD have the most general involvement. However, from a statistical point, MD brain regions have fuzzy boundaries, strongly dependent on the number of examined tasks, the (behavioural) variability across these, among other things. Applying a statistical threshold (i.e. conjunction analysis) is a crude (but effective) way to simplify this space and identify regions associated in **all** tasks. In contrast, if we follow Duncan and Fedorenko approach and use a Z statistic, we change our question to identify regions associated with **most** tasks. Alternatively, the Union mask reflects regions associated in **any** task. In our conjunction analysis we emphasise the fact (as the reviewer pointed earlier in P.2.1) that **most** voxels in the Union mask are active in **most** tasks. We also show that the

task pairwise spatial similarity derived from this method is strongly associated with the external psychometric similarity, suggesting a link between form and function. However, we agree with the reviewer that this is a crude approach and is susceptible to the same problem as images fallacy (as it is highly dependent on arbitrary statistical thresholds and ignores possible interactions). Therefore we now expand on this binary pairwise comparison and show that even within the 12-way intersection mask, containing voxels that are most commonly activated across tasks by definition, large variability is apparent across the group task activation patterns. This multivariate coding of tasks accords well with the sampling theory. It is also probably what enables the high classification accuracy of the tasks. We include these additional analyses and substantially rework the way in which we report and discuss the results to clarify the above.

Reviewer Point P 2.3 —” *”The key tenants of a “network sampling theory of human intelligence” could be more clearly laid out and contrasted with other theories of intelligence (maybe in a special panel/table/figure).”*

”

Reply: We have now reworked the text to clarify the main assumptions of the network sampling theory of human intelligence. We touch the comparison to previous theories in specific sections of the discussion and the introduction for lack of space.

Reviewer Point P 2.4 —” *I could not follow section 7 and Figure 4 of the results. What is being shown in Figure 4 A? The reasoning for trying to find the top 3 most-distinguishable tasks is not clear. Why 3? There is no comparison with other iterations of the 3-way classification shown, so the stability of these results is difficult to evaluate. Why not use the 3 assignments from (VS, Re, and Ve) for 3-way classification? Couldn't these results be driven by differences across tasks in stimulus features (e.g. visual and motor) introduced by the experimenters? ”*

Reply: We **really** thank the reviewer for this comment, we have now removed both the section and the figure from our revised paper. Furthermore, we did exactly as suggested and compare between the psychometric characteristics (as shown now in figure 3) and visual and motor characteristics (derived manually from in-depth comparison across tasks) as can be seen in Figure 5 in our revised text. We also address the issue of stability by comparing each of these three perspectives (i.e. psychometric, motor and visual) independently to proportionally balanced random assignments and conclude that both the psychometric and motor characteristics are significantly better than random assignments. In contrast visual features are not. Therefore, the dominant driving factors that allow us to classify the 12-tasks are probably driven by both motor and psychometric task characteristics and less so by visual stimuli features.

Reviewer Point P 2.5 —” *Data from an auditory-imagery task is referred to in section 1 of the results and in figure 1B. The source of this data is not clear and needs to be made explicit. More problematically, the inclusion of these results is objectionable, given that they appear to be from a distinct dataset from the present study. Are these studies well matched to the other task conditions? Why include these results and not others? Does this truly add anything to the manuscript? The claim that MD regions are not involved in “all” cognitive operations appears to rely quite heavily on inclusion of the auditory-imagery task, yet it appears inappropriate to include this result in the present study. ”*

Reply: The original reason for including this external finding was to emphasise the point that voxels that are active for ALL tasks and classically described as MD are in part related to the visual nature of this intelligence testing battery (and almost all other batteries). However, we now make this point by superimposing resting state networks over the 12-way intersection showing that it is composed of brain regions associated to visual and attention networks. Therefore to reduce confusion we removed this analysis from the current submission.

Reviewer Point P 2.6 —” *A recent paper [Siegel JS, Mitra A, Laumann TO, Seitzman BA, Raichle M, Corbetta M, Snyder AZ (2016) “Data Quality Influences Observed Links Between Functional Connectivity and Behavior”. Cereb Cortex. 1–11.<http://doi.org/10.1093/cercor/bhw253>] established that in-scanner*

*motion is strongly anticorrelated with individual differences in general intelligence, and that this produces spurious correlations with fMRI functional connectivity measures. This puts a large burden on all neuroimaging studies to firmly establish that their effects are not driven by motion artifacts. The present study states in the supplemental materials: “Nuisance experimental matrices were formed using motion parameter estimates (in an extended 24-parameter model) and motion outlier (spikes) events.” First, more details are needed to evaluate how likely these steps were to remove the effects of motion on the connectivity (and activity) estimates (e.g., the motion outlier threshold). Second, this is likely insufficient for removing motion/physiological artifacts that correlate with intelligence, as explained in this paper: [Power, Jonathan D., Mark Plitt, Stephen J. Gotts, Prantik Kundu, Valerie Voon, Peter A. Bandettini, and Alex Martin. (2018). “Ridding fMRI Data of Motion-Related Influences: Removal of Signals with Distinct Spatial and Physical Bases in Multiecho Data.” *Proceedings of the National Academy of Sciences* 115 (9): E2105–14. doi:10.1073/pnas.1720985115]. In particular, it would be important to regress out global signal (or use aCompCor) to reduce the effects of physiological artifacts on the results. ”*

Reply: Based on this comment we have changed the way we estimate functional connectivity, as described in short in the Dynamic functional connectivity section in the methods and in detail in the supplementary methods. Specifically, we regress out variables of no interest including: (1) FIR to capture activation, (2) tissues specific and global signal (and derivatives) (3) motion parameter estimates (including derivatives and squared transformation of these aka Friston 24-parameter model) (4) spikes (estimated using frame-wise displacement). Then using the residual we estimate functional connectivity measure using Pearson cross correlation. This new approach has no major impact on our main results or conclusions.

Reviewer Point P 2.7 —” *A series of papers have established that FIR regression (rather than canonical HRF, as is standard in PPI/gPPI analysis) should be used to remove the inflation effects of task activations on task functional connectivity estimates: [Cole MW, Ito T, Schultz D, Mill R, Chen R, Cocuzza C (2019) “Task activations produce spurious but systematic inflation of task functional connectivity estimates”. *NeuroImage*. 189:1–18. <http://doi.org/10.1016/j.neuroimage.2018.12.054>] and also [Al-Aidroos N, Said CP, Turk-Browne NB (2012) “Top-down attention switches coupling between low-level and high-level areas of human visual cortex.”. *Proceedings of the National Academy of Sciences*. 109:14675–14680. <http://doi.org/10.1073/pnas.1202095109>]. Without using FIR regression to remove the effects of task activations the task functional connectivity analyses are likely biased by the task activations, especially given that classification is used (which can pick up on subtle biases in the data). In particular, the task functional connectivity results cannot be trusted to be specific to connectivity, such that the task activation results might be re-described as “connectivity”. ”*

Reply: We have rerun the functional connectivity estimation using FIR as opposed to HRF convolved block trains. Changing over to FIR, as suggested, constituted a substantial amount of work requiring all of our analyses to be rerun. Importantly, this has not impacted our main findings.

Minor

Reviewer Point P 2.8 —” *The brain maps are small and of low quality, especially for figure 4 (where the orange and red are difficult to differentiate). It would be better to plot the brain maps with a higher-quality surface rendering to allow for better visualization of activity patterns. ”*

Reply: We appreciate this comment. We have improved both the resolution and smoothness of our maximum intensity plots. As this entire analysis is volumetric we feel that surface projections will give a false impression that this study relies on a surface analysis, therefore we hope the current improvements will be sufficient.

Reviewer Point P 2.9 —” *It’s unclear what Figure 1.C adds to the manuscript. It seems more beneficial to enlarge the other parts of the figure. ”*

Reply: We agree. The original intent of this figure was to emphasise the fact that connectivity measures relied on much less features than the voxel-wise counterparts. However, we took this criticism to heart and revised all of the figures leaving only important parts now. We also added this information in a table that highlights this relationship.

Reviewer Point P 2.10 —” *It is problematic that there is no Methods section in the manuscript. Many details are unclear despite some brief description of methods in the Results section. A detailed Methods section should be added to the main manuscript (not the supplement). Even in the supplemental methods there is not enough detail. In particular, many of the preprocessing parameters are not included, making it difficult to evaluate the quality of the analyses.* ”

Reply: We now include a methods section in the body of the paper in addition to a much detailed description and visualisation included in the supplementary material.

Reviewer Point P 2.11 —” *A more thorough discussion of MD cortex would be helpful for readers. The MD voxel map should be visualized in the main text and the ROI assignments shown in the supplement. (The brain parcellation in the supplement for WC could show the ROI mappings of the brain atlas used.)* ”

Reply: We now added a new analysis section with multi panel figure (Figure 2), making this point much clearer. In the figure we include an axial and sagittal projections of the different ROI sets. In addition, we have also included in the discussion a section on MD where we highlight previous findings and our view regarding MD cortex as a property of the dynamic functional connectome, as discussed in our response above.

Reviewer Point P 2.12 —” *Aspects of the manuscript feel unfocused, not fully fleshed out, and should be made more accessible. The last half of the discussion is especially confusing.* ”

Reply: The discussion has been substantially revised based on the new analyses and reviewers comments.

Reviewer Point P 2.13 —” *The observation that the so-called WM sub-systems of section 7 overlap “most densely” in MD cortex is not quantified (and/or no reference is provided). It is unclear if this is referring to the data-driven results in the present study or the results from previously published work.* ”

Reply: Following your previous comment (see P 2.8) we have removed both section 7 and figure 4 from the paper.

Reviewer Point P 2.14 —” *The correlations presented in figure 2 should probably be corrected for multiple comparisons.* ”

Reply: We are now reporting FDR corrected p-values for this results - see figure 6a-c .

Reviewer Point P 2.15 —” *Was the same null used for the classifications based on ROI, voxel, dFC, and stack? (Only one is shown in figure 1 D.)* ”

Reply: In the new analysis we removed ROI values to simplify the paper (and also because it is obvious that ROI data will be outperformed by the other metrics). To answer your question null approximation is calculated for each metric independently, as these are completely independent feature spaces. We don't approximate null for the stack model, as the object of the stack model is to examine whether combining these two BOLD metrics introduces a meta-classifier that out-performs each model independently. However, in the current visualisations in figure 4a,d,g we average the null models results for simplicity (after establishing no significant differences between them).

Reviewer Point P 2.16 —” *The authors indicated that 12 tasks comprised of 3 blocks of 1 minute duration (separated by 20 s of rest) were collected during BOLD fMRI scans with a TR of 2 s. In this case, it is*

perplexing how the authors examined 2160 events (60 time points x 12 tasks x 3 blocks), rather than 1080 events (30 time points x 12 tasks x 3 blocks). ”

Reply: This is partially answered in P 2.2, however, it is important to clarify that an event is defined as one 60sec block (which is 30 TRs) of participant engagement in one of our tasks. As we have 60 participants each performing 12 tasks for three times we have 2160 such events in total.

Reviewer Point P 2.17 —” *The author’s don’t explore the role of differences in how the tasks load onto ‘g’.* ”

Reply: That is correct - for this reason we have been careful not to refer to the behavioural composite measure as general intelligence. We do have measures from the Cattell for a sub-set of participants, and could potentially include this in a supplemental analysis. However, we do not think this is necessary for our core question/hypothesis.

Reviewer Point P 2.18 —” *There could be some nuance in distinguishing between the features of the brain that enable intelligent behavior at all and the features associated with high levels of intelligence.* ”

Reply: Using individual differences correlations, as we have, captures a substantial component of variability in intelligence. However, it does not capture all of the variance. Certainly it is hard to evaluate how well the classification-intelligence relationship holds at the extremes without running a much larger cohort, or specifically recruiting a very high intelligence group. This is actually a very interesting point that could be the focus of future studies.

Reviewer Point P 2.19 —” *It would be nice to comment on how this ties into the “efficiency” framework of intelligence (Schultz and Cole, 2016 [Schultz, Douglas H., and Michael W. Cole. (2016). “Higher Intelligence Is Associated with Less Task-Related Brain Network Reconfiguration.” *The Journal of Neuroscience* 36 (33): 8551–61. doi:10.1523/JNEUROSCI.0358-16.2016.]), which shows results essentially opposite of those presented here.* ”

Reply: We think that rest and task reconfiguration index suggested by Schultz and Cole is a useful way to collapse the connectome to one measure that has been shown to relate to Intelligence. Unfortunately, we didn’t record resting state sessions for these participants. However, we feel that the fact that participants with high performance index exhibit better overall task classification using dFC (but not BA) does not contradict these findings at all. In fact our new analysis (see Figure 6d and 6f,g) shows that the ability to decode a task from all other tasks is probably driven by identifying generalised sparse configuration of task-specific **between** resting state connections as well as consistent (and probably denser) task-general **within** resting state connections. It is highly likely that high performers will have more cohesive task-general connectivity in general as well as more efficient (i.e. sparser) task-specific connectivity. Therefore, in these participants the similarity to the resting state FC will be dominated by the dense task-general state and less affected by the sparse and effective task specific configuration.

Reviewer Point P 2.20 —” *Work on the figure legend descriptions. Make sure all axes labeled and legends (or descriptions in figure caption) are present in each figure. Define grey points (presumably null model) on Figure 3C. Work to clarify what each figure is showing, e.g. what points in scatter plots are (pairs of tasks vs individuals), abbreviations, etc. The text at the end of FFigure 4 caption seems like it belongs in the main text and the figure is missing a legend. Could say “classifier accuracy (F1-score)” in figures to ease accessibility.* ”

Reply: We went over all the figures and tried to include all relevant information on the figure legends. We hope we were able to make these clearer to the reader.

Reviewer Point P 2.21 —” *Edit more thoroughly. Semicolons are overused. There are lots of run-on sentences. There are typos and formatting issues throughout (especially in the supplement).* ”

Reply: We have thoroughly revised the manuscript and the supplementary material.

REVIEWERS' COMMENTS:

Reviewer #1 (Remarks to the Author):

This revised manuscript now reports a compelling, novel, and theoretically important test of the network sampling theory account of human intelligence. The authors were highly responsive to reviewer input, which resolved my previous concerns. The work is sophisticated and elegant, with multiple broad implications.

Reviewer #2 (Remarks to the Author):

The authors have done an excellent job addressing previous concerns. The re-analysis of the data using FIR task regression is especially impressive, given the amount of work involved but also how much it improved the results. It may have been difficult to appreciate this improvement because the main conclusions did not change, and yet the new results have a major set of confounds ruled out (improving confidence in the authors' interpretation of the results). The inclusion of task similarity based on separate perceptual, cognitive, and motor factors was also impressive and substantially improved the results (among other changes).